# DNA supercoiling-mediated G4/R-loop formation tunes transcription by controlling the access of RNA polymerase

Jihee Hwang[1,8], Chun-Ying Lee[1,8], Sumitabha Brahmachari[2], Shubham Tripathi[3], Tapas Paul[1], Huijin Lee[4], Alanna Craig [5], Taekjip Ha [1,4,5,6,7] & Sua Myong [1,4,5] ✉

RNA polymerase (RNAP) is a processive motor that modulates DNA super-coiling and reshapes DNA structures. The feedback loop between the DNA topology and transcription remains elusive. Here, we investigate the impact of potential G-quadruplex forming sequences (PQS) on transcription in response to DNA supercoiling. We find that supercoiled DNA increases transcription frequency 10-fold higher than relaxed DNA, which lead to an abrupt formation of G-quadruplex (G4) and R-loop structures. Moreover, the stable R-loop relieves topological strain, facilitated by G4 formation. The cooperative formation of G4/R-loop effectively alters the DNA topology around the promoter and suppresses transcriptional activity by impeding RNAP loading. These findings highlight negative supercoiling as a built-in spring that triggers a transcriptional burst followed by a rapid suppression upon G4/R-loop formation. This study sheds light on the intricate interplay between DNA topology and structural change in transcriptional regulation, with implications for understanding gene expression dynamics.

DNA topology and structure are dynamically modulated in response to gene expression. RNA polymerase (RNAP) generates supercoils in the DNA during transcription, facilitating or impeding successive transcription and other DNA-dependent biological processes[1]. Therefore, DNA could actively regulate transcription beyond simply serving as a passive information repository[2,3]. Normal B-DNA forms a right-handed double helix. However, negative supercoiling introduces strain into the DNA molecule, causing the double helix to twist and writhe to destabilize the hydrogen bonds and base stacking interactions between the base pairs, thus making it easier for the strands to separate. This physical feature of negative supercoiling can affect transcription in two ways: by regulating the RNAP loading rate and inducing noncanonical DNA

structures[4,5]. How such kinetic and mechanistic parameters are coupled to regulate transcription activity is poorly understood.

RNAP loading rate is susceptible to DNA supercoiling because a DNA duplex must open up at the promoter. Therefore, DNA's negatively supercoiled or underwound state facilitates the unwinding process of RNAP by lowering the energy barrier for strand separation[6,7]. In addition, the RNAP movement also dynamically generates and dissipates DNA supercoiling as described in the twin supercoiling domain model[8,9]. Adding to the complexity, negative supercoiling promotes the formation of noncanonical DNA structures[10,11], which further contributes to gene expression by modulating accessibility to transcription factors and RNAP[12,13]. However, it

[1]Programs in Cellular and Molecular Medicine, Boston Children's Hospital and Harvard Medical School, Boston, MA, USA. [2]Center for Theoretical Biological Physics, Rice University, Houston, TX, USA. [3]Yale Center for Systems and Engineering Immunology & Department of Immunobiology, Yale School of Medicine, New Haven, CT, USA. [4]Department of Biophysics and Biophysical Chemistry, Johns Hopkins University School of Medicine, Baltimore, MD, USA. [5]Biophysics, Johns Hopkins University, Baltimore, MD, USA. [6]Program in Cellular Molecular Developmental Biology, Johns Hopkins University, Baltimore, MD, USA. [7]Howard Hughes Medical Institute, Johns Hopkins University, Baltimore, MD, USA. [8]These authors contributed equally: Jihee Hwang, Chun-Ying Lee. ✉e-mail: sua.myong@childrens.harvard.edu

is difficult to fully understand what DNA structures are generated during transcription and how the resulting DNA topology regulates transcription. Thus, our study aims to investigate the impact of DNA topology and structure in regulating transcription.

G-quadruplex (G4) is a representative non-B DNA structure that can form during transcription. G4 arises in G-rich DNA sequences, which can fold into a four-stranded structure consisting of multiple stacks of G-quartets. ~370,000 potential quadruplex-forming sequences (PQS) have been predicted in the human genome and are highly enriched in the promoter region of regulatory genes, splice junctions, and the 5′ untranslated region (5′ UTR) of transcriptionally active genes, particularly in oncogene promoters[14–18]. Therefore, they are proposed to regulate the expression of these genes and are targeted by small molecules for therapeutic intervention[16,19,20].

Recent studies provided substantial in vivo evidence that G4 is present in the transcribed regions of the human genome[18,21–24]. However, in vitro assays showed that PQS forms a duplex over G4 under physiological conditions[25–27]. Thus, endogenous G4 formation requires strand separation to expose single-stranded DNA (ssDNA) from the duplex. Although negative supercoiling can facilitate the unwinding of the duplex DNA, it is insufficient to generate G4[25,28], because unlike other non-B DNA structures such as triplex (H-DNA), Z-DNA, and cruciform in plasmids[29–32], G4 requires more time to fold as each G quartet must be pre-folded and stacked[25]. To overcome the high kinetic barrier, G4 requires another way to stabilize the exposed ssDNA, such as forming other non-B structures in the complementary strand of PQS and the negative DNA torsional strain[33–35].

R-loop is a three-stranded structure that consists of a DNA:RNA hybrid and a complementary strand of displaced ssDNA[36]. R-loops are generated during transcription of G-rich sequences on the coding strand[37] and can extend up to several kilobases (kb) through high GC contents or GC skew[38–40]. R-loop facilitates the G-rich non-template strand to form into G4. Recently, two single-molecule studies directly visualized R-loop and G4 formation during transcription[41,42]. Negative supercoiling also promotes R-loop formation by facilitating strand separation, which is required for annealing nascent RNA transcript with the DNA template[43]. M. Drolet et al. have shown that negative supercoiling is the primary inducer of R-loop formation in vitro and in vivo[44,45]. Furthermore, DNA topoisomerase I, an enzyme that relaxes the supercoiled DNA by removing helical constraints, acts as a critical cellular regulator of R-loop homeostasis from bacteria to humans[45–47].

Previous studies suggest an intimate interplay between negative DNA supercoiling and G4/R-loop structures during transcription. In addition, the rate of G4/R-loop formation depends on the transcription efficiency[2,48,49]. However, how much, how quickly, and in what order they form, and how they impact transcription remains unclear. Our previous work found that co-transcriptionally formed R-loops induce G4 formation in non-template, which enhances transcription yield in linear DNA[41]. Here, we investigated how negatively supercoiled DNA impacts co-transcriptional R-loop and G4 formation and the corresponding transcription output. We applied ensemble fluorescence and gel-based assays to measure the mRNA production and concomitant structural changes in DNA during transcription. Unlike in linear DNA, the formation of G4/R-loop in supercoiled DNA reduces transcription position-dependent. Our newly developed single-molecule measurement, supplemented by a simulation model, reveals a new regulatory switch-like on-off mechanism that entails (i) switching on: negative superhelicity of DNA induces a dramatic transcriptional bursting with 10-fold higher frequency compared to a relaxed DNA (ii) switching off: transcriptional burst which accumulates negative supercoiling leads to an abrupt formation of G4/R-loop structures which absorbs negative superhelicity, thereby suppressing transcription. In summary, we provide new insights into G4/R-loop-mediated gene regulation in a DNA topological context. Furthermore, our innovative single-molecule

detection platform enables FRET measurement on a long plasmid DNA construct, reporting the mechanical coupling between the transcription activity and DNA conformation.

## Results

### Distinct transcriptional regulation by PQS in the 5′ UTR in supercoiled and linear DNA

To investigate the impact of potential quadruplex-forming sequence (PQS) on transcription in supercoiled DNA, we employed a real-time ensemble in vitro transcription assay in which a quenched molecular beacon fluoresces upon annealing to the transcribed RNA (Fig. 1a). We prepared a 3.7 kb plasmid, purified from *E. coli*, containing T7 promoter and PQS (Fig. 1a). We generated linearized DNA as a control by treating the plasmid with a single-cut restriction enzyme. The 18-nucleotide (nt) ssDNA molecular beacon with a sequence complementary to the transcribed RNA was end-modified with Cy3 and a black hole quencher (BHQ2) such that it only fluoresces upon annealing to the mRNA product (Fig. 1a)[41]. A real-time Cy3 signal, which represents mRNA production, was obtained using a plate reader. We first explored the effect of PQS orientation on mRNA production for both linear and supercoiled DNA templates (Figs. 1b and c). PQS was positioned 30 bp downstream of the transcription start site (TSS) in either the non-template (PQS-NT) or template (PQS-T) strand or a random non-PQS sequence was inserted as a control. The linear increase of the fluorescence for each construct was normalized against the control (see the method section). For linear DNA, consistent with our previous findings, we observed an enhanced mRNA production rate in PQS-NT, with an increase of approximately 30% compared to the control and about 40% compared to PQS-T, respectively (Fig. 1b and c). In contrast, PQS-NT in plasmid did not show an enhanced RNA production rate compared to control[39]. The different effects of PQS-NT in linear vs. supercoiled DNA underscores the importance of DNA topology in regulating transcription activity.

### Transcriptional suppression is PQS position-dependent

Next, we asked if the PQS-NT positions impact transcription. We prepared linearized and enclosed plasmid with PQS inserted at 10, 30, and 60 bp downstream of TSS in the non-template strand (NT) (NT-sp10, NT-sp30, NT-sp60) (Fig. 1d). Linearized DNAs with NT-sp10, -sp30, and -sp60 showed no significant difference in RNA production rate, indicating that PQS position does not affect transcription in linear DNA (Fig. 1d). However, the same set of PQS in plasmid exhibited notable differences in RNA production rate. In particular, NT-sp10 and NT-sp60 showed ~40% and ~20% reduced RNA production rate than NT-sp30, respectively. Further tests on the PQS-NT positions at 10, 15, 20, 25, 30, 35, 45, and 60 bp downstream of TSS in the supercoiled plasmid (Fig. 1e) displayed an interesting pattern i.e., the PQS located proximal to TSS (NS-sp10, 15, 20) displayed the highest level of transcription suppression, which is gradually alleviated downstream (NT-sp25, 30) and again suppressed further downstream (NT-sp35, 45, 60). Importantly, all constructs led to a lower rate of RNA production than the control. Taken together, PQS in supercoiled DNA suppresses transcription in a position-dependent manner, while the same PQS in linear DNA enhances transcription. Such contrast emphasizes the unique regulatory role of PQS in supercoiled DNA.

### PQS-driven DNA topological changes mediate transcription suppression through G4-stabilized R-loop formation

Since the PQS-mediated transcription suppression was exclusive to supercoiled DNA, we hypothesized that DNA undergoes PQS-driven topological changes during transcription. Two PQS positions, NT-sp10 and NT-sp30, were chosen based on the high and low levels of suppression, respectively (Fig. 1d and e). We performed in vitro transcription reactions with NT-sp10 and NT-sp30 in linear and supercoiled DNA and subsequently applied the transcribed samples on a 1% agarose gel. In

supercoiled DNA, both NT-sp10 and NT-sp30 underwent a substantial topological shift from primarily supercoiled to a relaxed or nicked state during transcription (Fig. 2b, left panel). In contrast, their linear counterparts remained in the same position (Fig. 2b, right panel). The cross-section analysis of the gel image reveals that the upward band shift is more pronounced in NT-sp10 than in NT-sp30 (Fig. 2c). Based on the higher suppression seen in NT-sp10, the shifted bands likely correspond to DNA conformations that suppress transcription.

Next, we tested if the observed band shift is due to R-loop formation, as PQS-NT is prone to lead to R-loop formation[41] and R-loops

alleviate torsional stress in DNA molecules[50]. When we treated the transcribed samples with RNase H, which explicitly degrades RNA in the R-loop, all the shifted bands from both NT-sp10 and NT-sp30 returned to the original, i.e., primarily negatively supercoiled state of the plasmid (Fig. 2b and c, lanes 1 and 3). The electrophoretic mobility shift assay with S9.6, a monoclonal antibody specific to R-loops, further confirmed that all the shifted bands consisted of R-loops (Supplementary Fig 1). These results indicate that (i) R-loop formation is responsible for the observed band shift for both NT-sp10 and NT-sp30, (ii) the NT-sp10 induces a higher level of R-loop than in NT-sp30, and

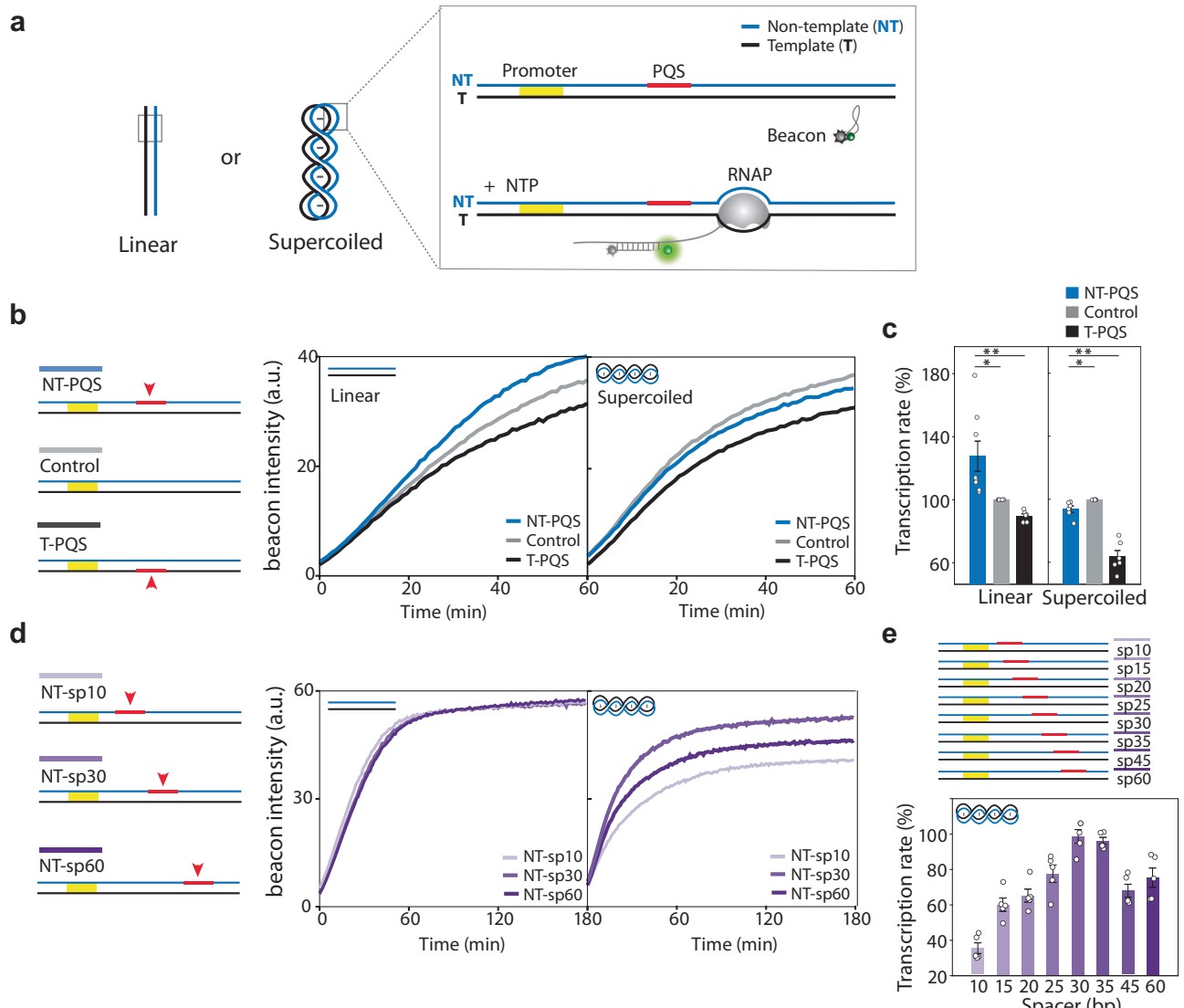

**Fig. 1 | Impact of DNA Supercoiling and PQS Position on Transcription Rates Measured by Beacon Assay. a** The schematic of beacon assay. Linearized and enclosed plasmids with identical sequence contexts are used to test the impact of DNA supercoiling on transcription. The DNA templates contain the T7 promoter and PQS, colored yellow and red, respectively. RNAP, colored gray, reads the DNA template and produces RNA, which anneals with the beacon, causing Cy3 signal enhancement. Real-time Cy3 signal measurements are obtained using a plate reader. **b, c** PQS orientation effect on RNA production. PQS locates 30 bp downstream of the Transcription start site (TSS) either in non-template (NT-PQS) or in template (T-PQS). As a control, random sequences are inserted (Control).
**b** Linearized plasmid with NT- PQS (left, blue line) shows an enhanced RNA production rate, while T-PQS (left, black line) shows a decreased RNA production rate compared to Control (left, gray line). The plasmid with NT-PQS (right, blue line) does not exhibit enhanced RNA production compared to Control (right, gray line)

and T-PQS (right, black line). **c** Initial RNA production rate is calculated from the early linear part of the curve in (**b**). Cy3 signals are normalized by that of each Control. Data are presented as mean ± SEM of 6 - 8 replicates of experiments. The exact data and *P*-values are provided in the Source data file. *$P < 0.05$, **$P < 0.005$ (two-sided unpaired *t* test) (**d**) PQS position effect on transcription. PQS positioned at 10, 30 and 60 bp downstream of TSS in non-template (NT-sp10, NT-sp30, NT-sp60). All linearized DNAs with NT-sp10, NT-sp30, and NT-sp60 are dashed faint, light, and dark purple lines, respectively. All plasmids with NT-sp10, NT-sp30, and NT-sp60 are solid faint, light, and dark purple lines, respectively. **e** Normalized Cy3 signals for -sp10, -sp15, -sp20, -sp25, -sp30, -sp35, -sp45 and -sp60 of NT-plasmid. The signals are normalized by the signal of Control. Data are presented as mean ± SEM of 5 replicates of experiments. The exact mean ± SEM and *P*-values are provided in the Source data file.

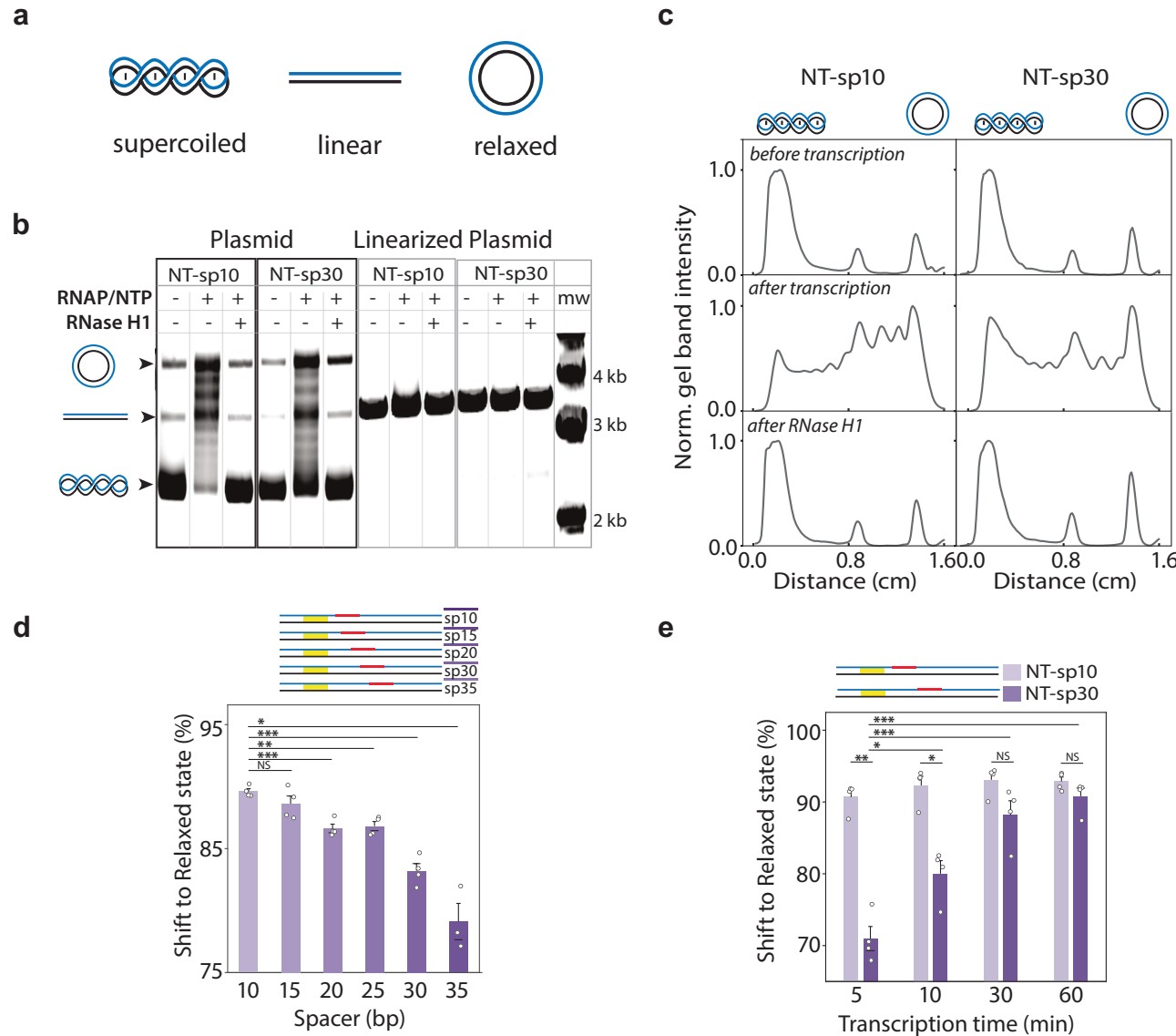

**Fig. 2 | PQS-Dependent Topological Shifts and R-loop Formation Regulating Transcription. a** Schematic of DNA supercoiling status. **b** Gel retardation assay for in vitro transcription experiments with plasmid-NT-sp10, -NT-sp30 and linearized counterparts (lane 2, 5, 8, and 11). To address the components of the shifted bands, RNase H, an enzyme to degrade RNA in R-loops, is treated in the transcribed samples (lanes 3, 6, 9, and 12). In plasmids, NT-sp10 and NT-sp30 (lanes 2 and 5) exhibit a strong topological shift where the bands reach the partially or fully relaxed substrate position. RNase H treatment fully returns the shifted bands of plasmid-NT-sp10 and -NT-sp30 (lanes 3 and 6) to their initial negatively coiled DNA state (lanes 1 and 4), indicating that R-loop formation accompanies the topological shift.

In contrast, linearized NT-sp10 and NT-sp30 show no heterogenic structures on the gel (lanes 8, 9, 11, and 12). **c** Intensity plot of the gel bands for plasmid-NT-sp10 and -NT-sp30 from (**b**) reveals different degrees of shiftiness in-band distribution. **d** Quantification of the band shiftiness for transcribed samples of plasmid-NT-sp10, -sp15, -sp20, -sp25, -sp30, and -sp35 at a fixed transcription time (30 min). **e** Quantification of the band shiftiness over transcription time for plasmid-NT-sp10 and -NT-sp30. For (**d**) and (**e**), data are presented as mean ± SEM of 3 - 4 replicates of experiments. The exact data and *P*-values are provided in the Source data file. *$P < 0.05$, **$P < 0.005$, ***$P < 0.0005$, NS: nonsignificant (two-sided unpaired *t* test).

(iii) a higher R-loop in NT-10 drive stronger transcriptional repression. The R-loop is an inhibitory structure for transcription under supercoiled DNA conditions. This relationship is demonstrated by the gradient of R-loop forming propensity: highest to lowest R-loops are detected for NT-sp10, -sp15, -sp20, -sp30, -sp35 (Fig. 2d), which is inversely related to the transcription rate (Fig. 1e).

Furthermore, we measured the kinetics of R-loop formation in NT-sp10 and NT-sp30 by taking samples at 10, 20, 30, and 60 min of transcription reaction and analyzing them by the gel shift assay (Supplementary Fig 2). While NT-sp10 displayed a rapid topological shift within the first 5 min, NT-sp30 showed a more gradual shift over 60 min (Fig. 2e). The faster shift seen in NT-sp10 than in NT-sp30 indicates that the early-stage R-loop formation leads to a more drastic reduction in

transcription in NT-sp10 (Fig. 2f). To test if G4 contributes to the R-loop mediated transcriptional suppression, we applied G4 destabilizing (no mono-valent ion, LiCl) and stabilizing (KCl) conditions (Supplementary Fig. 3)[51,52]. The G4 stabilizing condition led to increased topological relaxation only for NT-sp30 but not for control or T-sp30, suggesting that the formation of G4 structures in NT-sp30 promotes topological shifts, i.e., R-loop formation. These findings further support that G4 forms and plays a pivotal role in stabilizing R-loop structures.

## DNA Supercoiling modulates the impact of PQS on transcription regulation

Our results suggest that G4/R-loop structures inhibit transcription primarily when PQS is located near the TSS in the non-template strand

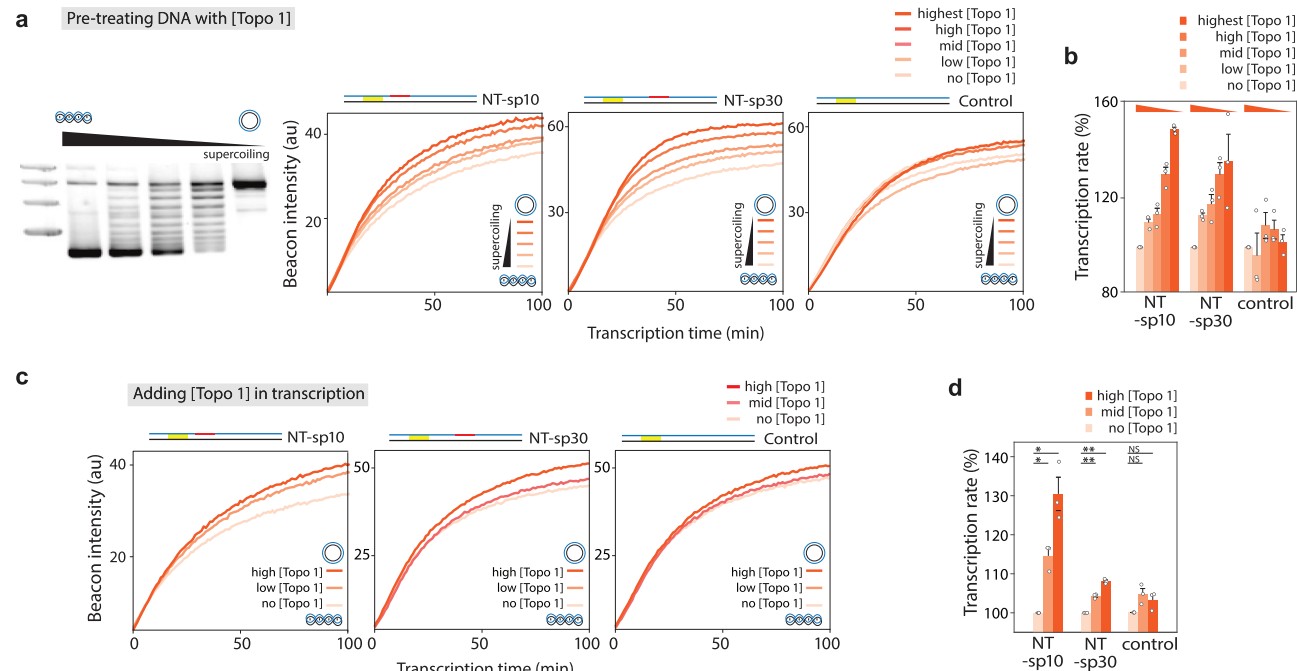

**Fig. 3 | The impact of PQS in transcription regulation diminishes with reduced DNA supercoiling. a** Beacon assay (described in Fig. 1) conducted with DNA templates in various topological states. Plasmid-NT-sp10, NT-sp30, and control samples were pre-treated with topoisomerase I (Topo I) at different concentrations to create varying degrees of superhelicity. The gel image on the left shows NT-sp10 treated with 10 units (highest), 5 units (high), 3.3 units (mid), and 2.5 units (low) of Topo I, representing a gradient of supercoiling levels. For further details, see Supplementary Fig. 4 and the Methods section. In vitro transcription reactions are conducted with the pre-treating DNAs with Topo 1 (right). The more orange-colored the line, the less supercoiled the DNA construct is. (b) Bar graph of (**a**) displaying the normalized Cy3 signal increase for each DNA construct. Bars represent the mean ± SEM from three independent experiments. **c** In vitro transcription reactions are conducted with increasing concentrations of Topo 1, which gradually relaxes negative supercoiling during transcription. **d** Bar graph of (**d**) displaying the normalized Cy3 signal increase for each DNA construct. For (**b**) and (**d**), data are presented as mean ± SEM of 3 - 4 replicates of experiments. The exact data and *P*-values are provided in the Source data file. *$P < 0.05$, **$P < 0.005$, ***$P < 0.0005$, NS: nonsignificant (two-sided unpaired *t*- test).

of negatively supercoiled DNA. In cells, such suppression can be mitigated by the activity of topoisomerases, which can relax torsional strains built into DNA. Therefore, we asked if topoisomerase-induced relaxation of the supercoiled DNA could reverse the inhibitory effect of the G4/R-loop structures. We investigated the relaxation effect by treating the DNA with topoisomerase I (Topo I) prior to transcription (Fig. 3a and Supplementary Fig. 4) and by adding Topo I during transcription. For pre-treatment, plasmids were incubated with varying units of Topo I (10, 5, 3.3, and 2.5 units) for 30 min at 37 °C, followed by purification to control the degree of relaxation. Gel analysis confirmed the resulting gradient of superhelicity levels, as shown in Supplementary Fig. 4. Pre-treating DNA with Topo 1 showed a dose-dependent increase in transcription in the NT-sp10 and NT-sp30 (Fig. 3a and b), while the change was less evident in the control. Transcription performed in the presence of Topo I also showed enhanced transcription in a Topo 1 dose-dependent manner as Topo 1 progressively resolves negative supercoiling during the transcription (Fig. 3c and b). Both results demonstrate that the level of negative supercoiling is highly correlated with the suppressive effect imparted by the G4/R-loop structures, and such suppression can be lifted by Topo 1 (Fig. 3e).

### Real-time single-molecule detection of co-transcriptional R-loop and G4 formation

We employed single-molecule assays to probe the real-time dynamics underlying DNA conformational changes and topological shifts during transcription. We employed single-molecule (sm) FRET to measure DNA structural change[53,54]. and the single-molecule protein-induced fluorescence enhancement (smPIFE) to measure RNAP binding[55,56]. We constructed linearized, relaxed, and negatively supercoiled 3.7 kb

plasmid DNAs with two types of labeling strategies: [FRET1] for monitoring G4/R-loop formation and RNAP movement and [FRET2] for monitoring the transcription initiation process (Fig. 4a). Using a nicking enzyme-based internal labeling method[57], Cy3, Cy5, and biotin-labeled oligomers were annealed to the 3.7 kb-sized vectors and sealed through ligation (Fig. 4b). Negatively supercoiled DNA was prepared by exposing the annealed DNA to ethidium bromide during the ligation process (Fig. 4b left, lane 1). Without ethidium bromide, it generates relaxed DNA (Fig. 4b left, lane 2). As shown in Fig. 4b left, we imaged the DNA constructs on 1% agarose gel without post-staining because the constructs were already labeled with Cy3 and Cy5 fluorophores. By post-staining the gel, we confirmed that the Cy3 and Cy5 labeled supercoiled DNA exhibited a similar level of superhelicity with the plasmid purified from *E. coli* (Fig. 4b, right). We verified that the Cy3 and Cy5 incorporated into 3.7 kb plasmid were successfully immobilized on the surface and showed the expected FRET values (E) of ~ 0.3 for both constructs (Fig. 4c). The linearized DNA was prepared by cutting the relaxed DNA with a single-cut restriction enzyme (Supplementary Fig. 5).

### Transient R-loops promote G4/R-loop formation

[FRET1] has Cy3 and Cy5 across NT-sp30, labeled at 27 bp (+ 27) and 47 bp (+ 47) downstream of TSS, respectively. Based on our previous study, the expected FRET values for the DNA-only, DNA with R-loop, and DNA with G4 are ~ 0.3, ~ 0.7, and ~ 0.9, respectively[41]. In addition, we expect to visualize RNAP elongation near Cy3 at +27 via the PIFE signal (Fig. 5a).

For the relaxed [FRET1] construct, we observed primarily short-lived PIFE peaks without FRET change, which reports on successive RNAP elongation without any structural changes in the DNA

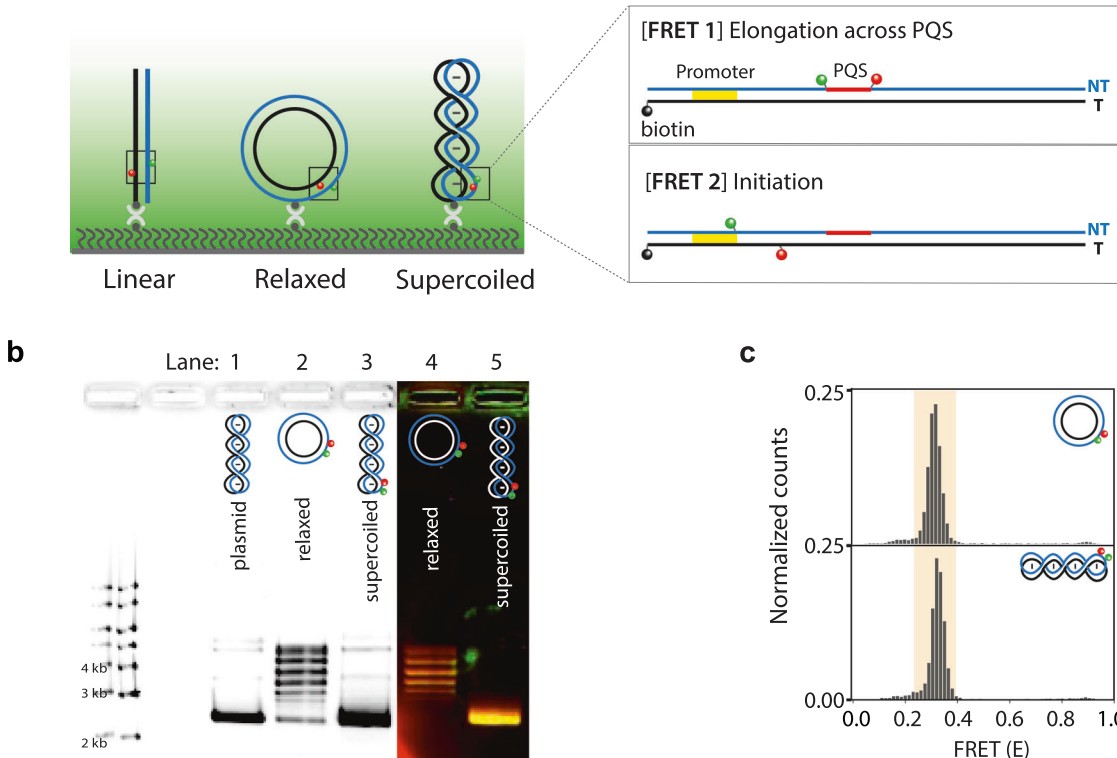

**Fig. 4 | Single-Molecule Detection Platform for Observing DNA Structural Changes and RNA Polymerase Movement on Immobilized Linearized, Relaxed, or Highly Negatively Coiled 3.7 kb DNA. a** The schematic representation of immobilized 3.7 kb DNA with linearized, relaxed, or highly negatively coiled DNA templates, which are specifically labeled with Cy3 and Cy5. Each DNA construct was immobilized on a PEG-coated glass slide via biotin-neutravidin interaction for single-molecule observation. To simultaneously observe the structural configuration of G4 and R-loop, and RNAP movement, Cy3 (+27) and Cy5 (+47) were positioned across the PQS [FRET1]. To monitor the transcription initiation process, Cy3 (−4) and Cy5 (+19) were positioned close to the promoter [FRET2]. **b** Using a nicking enzyme-based internal labeling method, Cy3, Cy5, and biotin-labeled

oligomers were annealed with the 3.7 kb-sized vector and sealed through ligation. During ligation, the presence or absence of EtBr determined whether the construct was relaxed or supercoiled. See the method section for details. The reconstituted constructs were confirmed with 1% agarose gel electrophoresis. The post-stained image with SYBR Green II displays a plasmid, the original template purified from *E. coli* (lane 1), along with Cy3 and Cy5 labeled relaxed (lane 2) and supercoiled DNA (lane 3). The fluorescence image without post-staining (lanes 5 and 6) displays Cy3 and Cy5 labeled relaxed DNA (lane 5) and supercoiled DNA (lane 6). **c** Histograms of relaxed and supercoiled [FRET1] DNA measured by single-molecule TIRF, displaying ~0.3 FRET value as expected.

(Fig. 5b)[56–58]. The short-lived PIFE peaks indicate elongating RNAP as the peak frequency increased as a function of RNAP concentration (Supplementary Fig. 6). However, no G4 and R-loop formations were observed in the relaxed construct within the first 150 s of observation time. In contrast, the supercoiled [FRET1] construct displayed two distinct patterns of FRET and PIFE signals: First, short-lived and frequent PIFE-FRET (E ~ 0.7) tandem peaks appeared (Fig. 5c and d). The time delay between PIFE and FRET peaks indicates that the PQS-bearing DNA segment undergoes a brief R-loop formation when RNAP transcribes through the PQS region. Second, after multiple rounds of short-lived PIFE-FRET transitions, the mid-FRET (E ~ 0.7) converts to a high-FRET state (E ~ 0.9) (Fig. 5e), signifying the transient R-loop to G4 transition respectively (Fig. 5c and e). Once it transitions to high-FRET, the state remains stable. The FRET histogram analysis revealed that the high FRET peak diminished under G4-destabilizing conditions by omitting KCl from the reaction buffer (Fig. 5f, no M⁺), confirming the long-lived high-FRET as a stably folded G4 state (Fig. 5f). The persistence of these high-FRET states indicates that no conformational change occurs within the G4. The RNase H treatment led to the disappearance of the mid-FRET (~ 0.7) peak, as expected from the R-loop removal. The 0.7 FRET peak is not distinct because the R-looped state is highly transient in supercoiled DNA, as demonstrated in Fig. 5c–e. In addition, the R-loop exhibited a broad range of FRET values, reflecting the transient and fluctuating nature of R-loop formation. Remarkably,

97.5% (119/122) of molecules that folded into G4 (E ~ 0.9) exhibited stepwise FRET transition from E ~ 0.7 to E ~ 0.9 (See more representative traces in Supplementary Fig. 7). This observation implies that the transiently formed R-loop is required to form a stable G4 structure. The transiently formed R-loop likely creates a locally unwound region, which may reduce the energetic barrier for PQS to fold into G4. In addition, supercoiled [FRET1] didn't show a long-lived R-looped state, unlike linear [FRET1] (Supplementary Fig. 8). The gel retardation assay showed that G4/R-loop induces topological relaxation, which contributes to transcriptional suppression (Fig. 2 and Supplementary Figs. 2, 3). Based on the result, the stable high FRET state likely encompasses both R-loop and G4 (G4/R-loop).

## Supercoiling accelerates RNAP loading, which stimulates G4/R-loop formation

One striking difference is that the supercoiled DNA construct exhibited a significantly higher frequency of PIFE spikes than the relaxed construct before forming the stable high-FRET state (E ~ 0.9) (Fig. 5d). When we quantified the number of PIFE peaks (Fig. 5g), supercoiled DNA showed an approximately ten-fold increase in transcription rate compared to the linearized and relaxed DNA. We postulated that the increased transcription rate that involves frequent RNAP loading would lead to the underwinding of the upstream DNA, which may increase the probability of G4/R-loop formation. As expected,

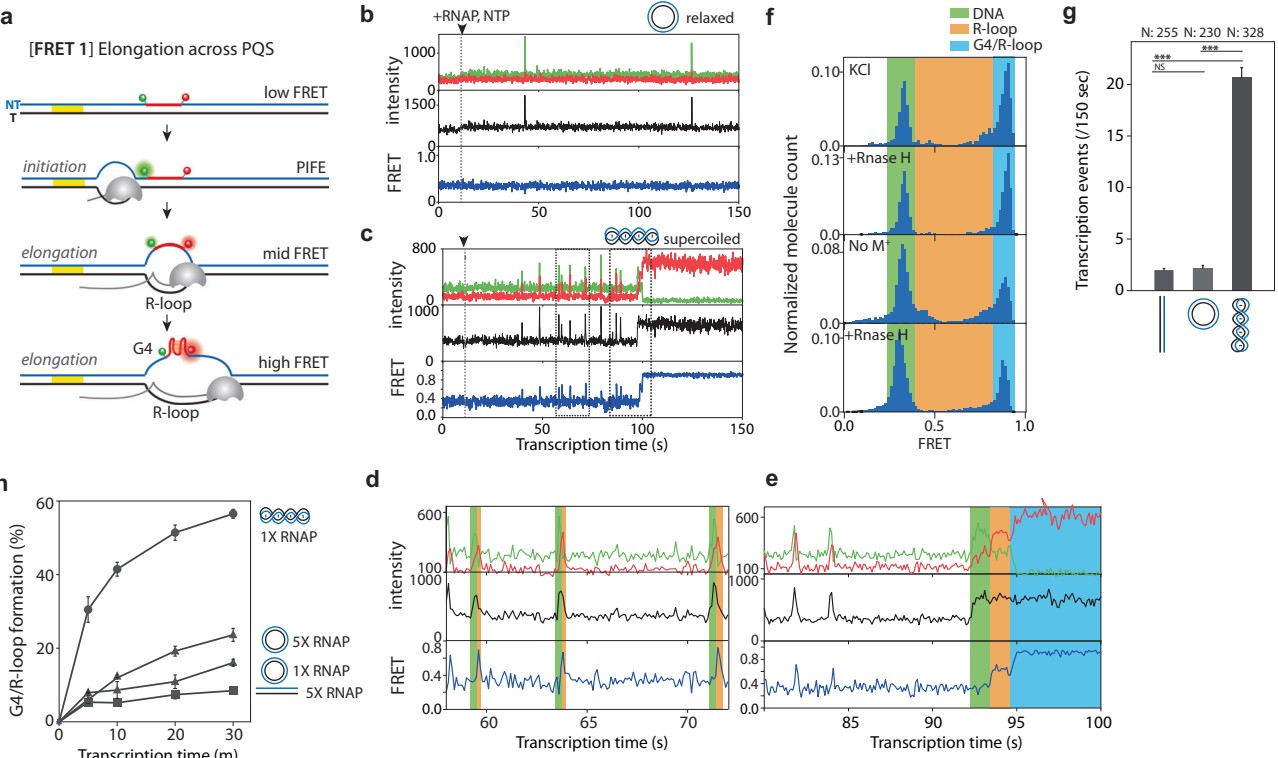

**Fig. 5 | Single-molecule investigation of DNA structural changes and transcription dynamics affected by DNA superhelicity. a** Schematic illustration of [FRET1] designed for direct observation of elongating RNAP and DNA structural changes. PQS is positioned 30 bp downstream of TSS in the non-template strand (NT-sp30), with Cy3 located 27 bp (+ 27) and Cy5 located 47 bp (+ 47) downstream from TSS. Linearized, relaxed, and supercoiled [FRET1] constructs were generated to investigate the impact of DNA superhelicity on RNAP movement and DNA structural modifications. In the absence of any event, the construct exhibits an expected FRET value (E) of ~ 0.3. As elongating RNAP approaches Cy3, the E remains at ~ 0.3, while the Cy3 signal is enhanced due to Protein-Induced Fluorescence Enhancement (PIFE). The expected E of R-loop and G4 is ~ 0.7 and ~ 0.9, respectively. **b, c** Representative single-molecule time traces of RNAP transcription for relaxed (**b**) and supercoiled (**c**) [FRET1] DNA constructs. The top graph shows Cy3 (green) and Cy5 (red) signals; the middle graph shows the combined intensity of Cy3 and Cy5, illustrating PIFE occurrence by RNAP passing Cy3; the bottom graph shows FRET changes, indicative of DNA structural alterations. The dashed gray line marks the time point when RNAP is introduced into the channel. **b** With the relaxed [FRET1], no FRET changes are observed, but short-lived PIFE signals are present. **c** With the supercoiled [FRET1], FRET transitions between ~ 0.3 and ~ 0.7 occur with short-lived PIFE peaks (**d**) A closer examination of the dashed black box on the left side of (**c**) reveals a distinct temporal shift between PIFE and FRET peaks, with a

FRET peak emerging shortly after a PIFE peak. **e** The dashed black box on the right side of (**c**) shows that after several transitions, an additional FRET change from ~ 0.6 to ~ 0.9 is observed. **f** FRET histogram of supercoiled [FRET1] after 30 min transcription (first from the top) and subsequent RNase H treatment (second). The orange boxed area corresponds to R-loop structures, which disappear in the presence of RNase H. The blue boxed area corresponds to stable G4 structures, which remain unaffected by RNase H treatment. FRET histogram of supercoiled [FRET1] after 30 min transcription in G4 destabilization condition (third from the top) and subsequent RNase H treatment (forth). The fraction of red boxed area decreases, corresponding to G4 destabilization and the blue box disappears in RNase H treatment, further confirming the state of R-loop and G4. **g** The quantification of PIFE peaks before forming stable G4 (E ~ 0.9) for linearized, relaxed, and supercoiled [FRET1] with more than 200 molecules for each experiment. See details in the method section. The exact data and *P*-values are provided in the Source data file. ***P < 0.0005, NS: nonsignificant (two-sided unpaired *t* test) (**h**) Time course of G4/R-loop formation affected by DNA superhelicity and RNAP concentration. The impact of DNA superhelicity on G4/R-loop formation rate in the following order: supercoiled [FRET1] with 1X RNAP (circle) > relaxed [FRET1] with 5X RNAP (triangle) > relaxed [FRET1] with 1X RNAP (triangle) > linearized [FRET1] with 5X RNAP (square).

supercoiled DNA exhibited a significantly higher fraction of G4/R-loop (high FRET) than relaxed DNA, even with a 5-fold higher RNAP concentration (Fig. 5h and Supplementary Fig. 8). This result is consistent with the shifted gel bands corresponding to the R-loop containing plasmid induced by the transcription activity (Fig. 2b). Thus, the negative supercoiling generated by the frequent transcription facilitates the quick and robust formation of G4/R-loop in supercoiled DNA but not in the relaxed or linear DNA. Therefore, supercoiled DNA is poised to induce rapid firing of transcription, which in turn triggers G4/R-loop formation that suppresses transcription. Furthermore, we confirmed that transcription-generated supercoiling also stimulates stable G4/R-loop formation by comparing relaxed DNA with linear DNA (Fig. 5h, 5X RNAP with relaxed DNA vs 5X RNAP with linear DNA). These findings suggest that supercoiling stimulates the formation of stable G4/R-loops from both pre-existing and transcription-generated DNA superhelicity.

## G4 formation stabilizes the R-loop state

To understand the mechanism by which G4 stabilizes the R-loop structure, we compared the transcription activity of NT-PQS vs. control non-PQS (Fig. 6a, b) in supercoiled DNA. The overall FRET pattern that emerges from both constructs was compared by generating a heatmap that consists of overlayed single molecule traces that are synchronized either at the time of RNAP + NTP addition (Fig. 6c, d) or post-synchronized at the time of long-lived G4/R-loop or R-loop formation (Fig. 6e, f). While transcription in NT-PQS led to a stable high-FRET state, transcription of a control DNA-induced oscillating mid-FRET state indicated an unstable R-loop formation without G4 (Fig. 6c, d). By comparing the time FRET changes occur, we find that NT-PQS DNA accelerates the transition into a stable FRET state. The second set of heatmaps for NT-PQS displays a clear two-step transition from low to mid to high FRET state, consistent with the single molecule trace shown

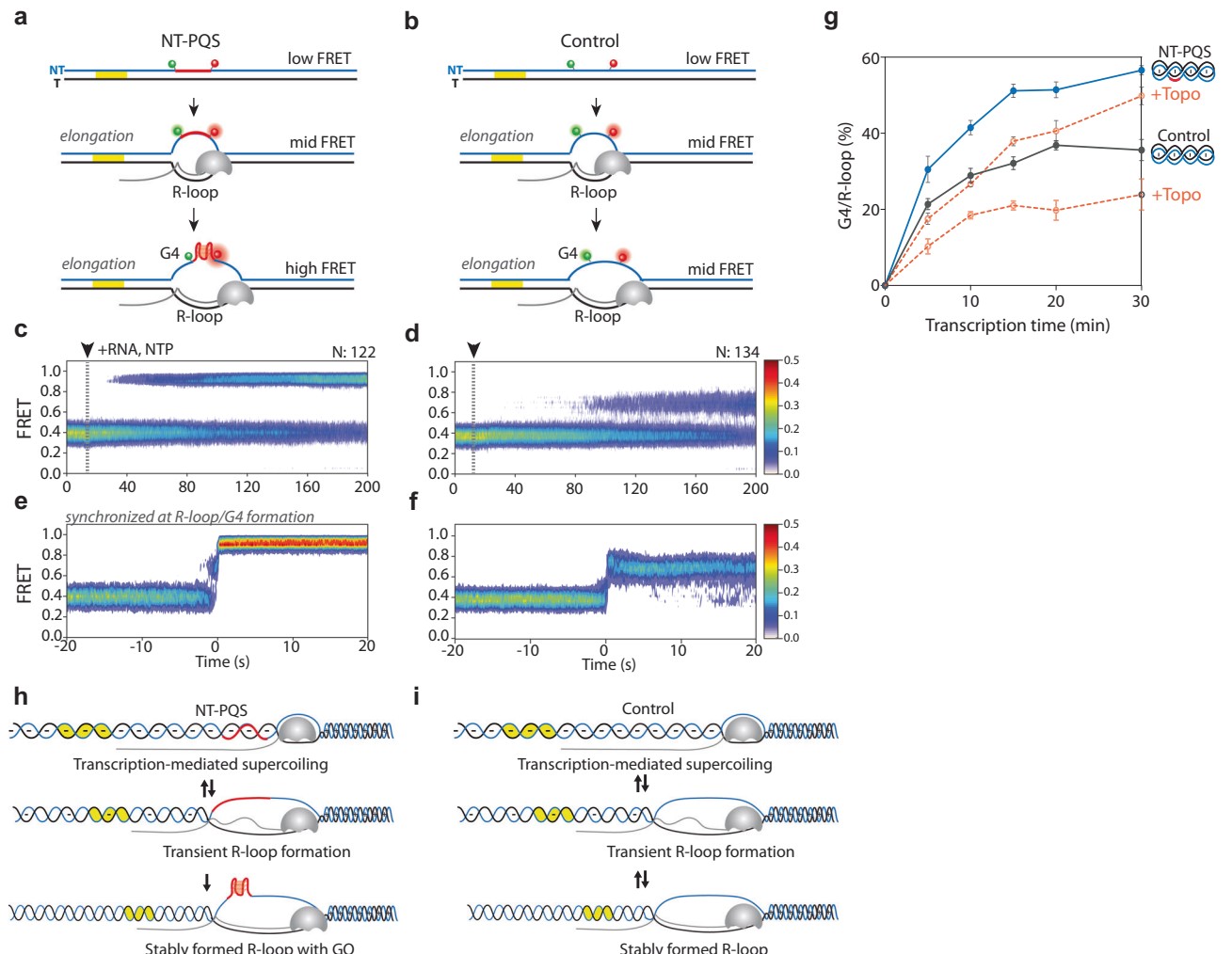

**Fig. 6 | PQS-Dependent Stabilization of R-loop Structure in Supercoiled DNA.**
**a–f** Heatmaps generated by overlaying traces that show transcription-induced DNA structural changes in supercoiled DNA with (**a**) NT-PQS (-sp30) and (**b**) control. **c–f** Single-molecule traces are synchronized either at the time of RNAP and NTP addition for NT-PQS (-sp30) (**c**) and control (**d**) or the time of forming long-lived G4 and R-loop structures for NT-PQS (-sp30) (**e**) and control (**f**). **g** Quantification of transcription-induced G4/R-loop or R-loop formation for NT-PQS (blue solid line and circle) and control (gray solid line and circle) in the presence of Topoisomerase I (orange dashed line and circle) (**f** and **g**) Schematic illustrations of reaction dynamics in the presence (**h**) or absence (**i**) of PQS in supercoiled DNA, emphasizing the role of PQS in driving the reaction to stabilize R-loop structures in supercoiled DNA.

in Fig. 5e. The control, which forms only R-loops, exhibits a fluctuating FRET signal before transitioning to a stable R-loop state or returning to the duplexed DNA (See the representative single-molecule traces in Supplementary Fig. 10). These observations demonstrate a unique feature of NT-PQS that enables G4 formation, stabilizing the R-loop state (Fig. 6h). Consistently, under G4 destabilizing condition (-M$^+$) (Supplementary Fig. 11), NT-PQS exhibits a lower R-loop formation rate and fluctuating FRET transitions, similar to the unstable R-looped state seen in control. These results further emphasize the role of G4 in stabilizing the G4/R-loop structure. Furthermore, treatment of Topo 1 decreased R-loop formation for both constructs, signifying the effect of supercoiling in R-loop stabilization regardless of the G4 formation (Fig. 6g).

## G4 and R-loop diminish the transcriptional burst

Next, we investigated the impact of G4/R-loop structures in transcription initiation. Based on the strand relaxation induced by R-loops, which suppress transcription (Fig. 2), we hypothesized that the stably formed R-loops and G4/R-loops impede RNAP loading onto the promoter. To test this, we adopted our previous FRET configuration suited for monitoring transcription initiation (Fig. 7)[41,58,59]. [FRET2] construct has Cy3 and Cy5 at 4 bp upstream (− 4) and 19 bp downstream (+19) from TSS, respectively. The expected FRET value of the immobilized DNA is ~ 0.4, which transitions to ~ 0.7 peak when RNAP opens the transcription bubble and transcribes> 4 nt. (Fig. 7a). Single-molecule traces showed frequent FRET peaks for varying durations (Fig. 7b). Although FRET 2 does not report on the formation of G4/R-loop, we interpret the sudden reduction in initiation events as coincident with the abrupt G4/R-loop formation shown in Fig. 5c–e. The emerging picture is that RNAP loading is accelerated due to the facilitated underwinding of the supercoiled DNA. This will promote R-loop and G4 formation, removing the supercoiling, thereby reducing the RNAP loading. To test this expected sequence of events, we quantified the FRET peaks at various stages of transcription over a 200 s interval at each time point—specifically at 0, 10, and 30 min after the addition of RNAP and NTP (Fig. 7c). It shows that transcription initiation is correlated with the formation of G4/R-loop (or R-loop in control) (Fig. 7c). Compared to the control, the NT-PQS exhibits faster G4/R-loop formation, leading to quicker suppression of transcription initiation. The level of negative supercoiling is likely reduced upon the formation of G4/R-loop structures to a level similar to that of a relaxed DNA, which

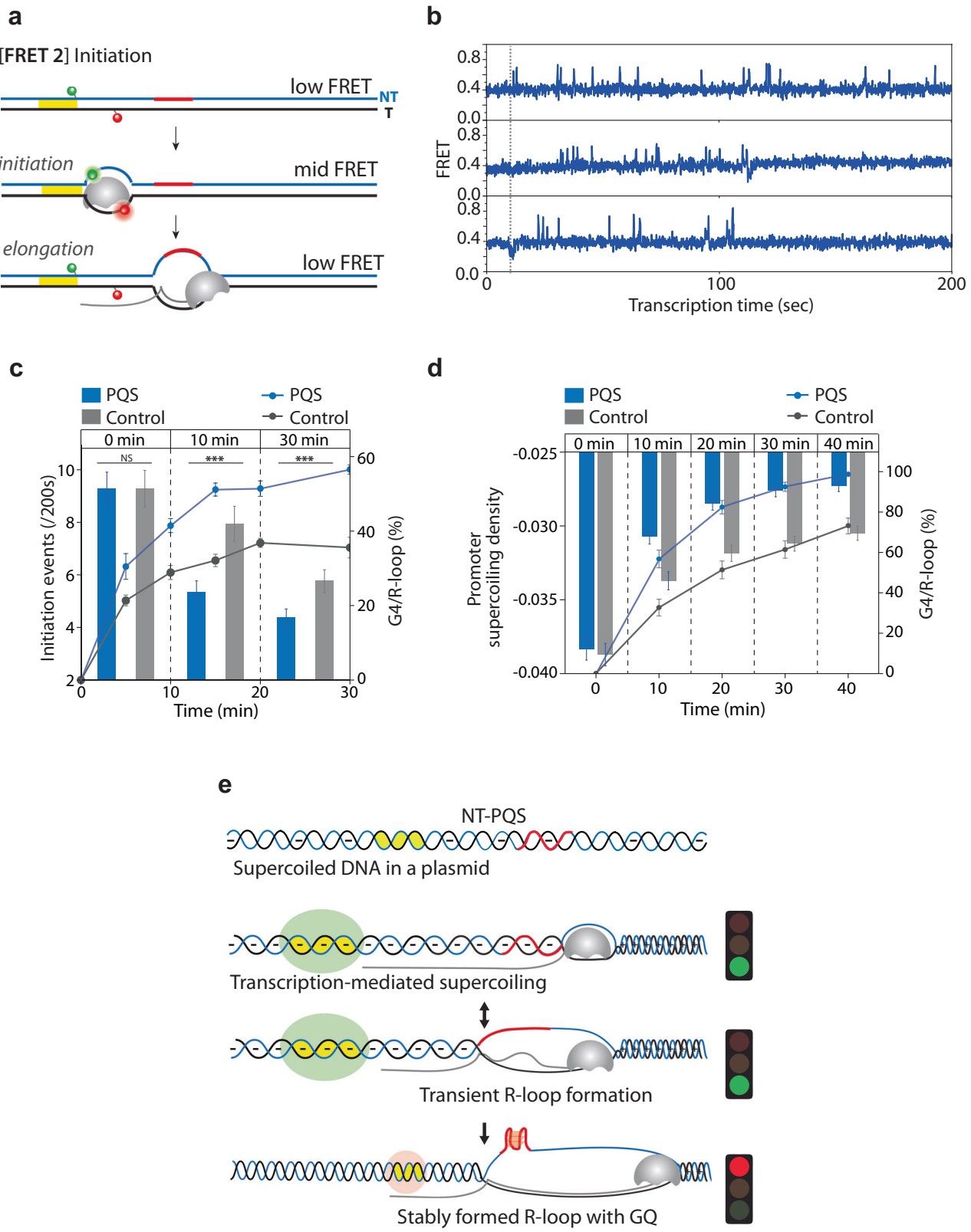

displays a markedly lower initiation frequency, averaging only around 3 events per 200 s (Supplementary Fig. 12). In addition, we employed an event-driven stochastic simulation to model supercoiling density changes in our system[60]. This simulation, as detailed in Tripathi et al. (2021), tracks the effects of co-transcribing RNA polymerases on DNA topology, specifically how transcription-induced supercoiling accumulates and influences the promoter region over time[60]. Through this

model, we observed that as G4/R-loops (or R-loops in the control) accumulate, the level of promoter supercoiling progressively increases. Notably, the NT-PQS construct shows a more rapid increase in supercoiling at the promoter compared to the control, as depicted in Fig. 7d and Supplementary Figs. 14 and 15. This model allows us to directly link the transcription-driven supercoiling dynamics with the regulatory effects of G4/R-loop structures, providing insight into the

**Fig. 7 | The formation of G4/R-loop structures decrease the transcription initiation rate in supercoiled DNA by absorbing the negative supercoiling near the promoter. a** Schematic illustration of [FRET2] for the direct observation of the transcription initiation process, featuring Cy3 and Cy5 positioned 4 bp upstream (−4) and 19 bp downstream (+19) from the TSS, respectively. The expected E of DNA is ~ 0.4. Upon transscription initiation, the E is ~ 0.7. When RNAP escapes from the promoter, the expected E is ~ 0.4. **b** Representative FRET traces of [FRET2] with PQS in non-template. A dashed gray line represents the time of RNAP and NTP addition. **c** Quantification of FRET peaks observed in 200 s after RNAP injection (0 min), 10 min, and 30 min for NT-PQS and Control (left *y*-axis with the blue bar and gray bar, respectively). Data points are quantified from 3–4 repetitions with over 200 molecules. Bars represent the average number of events with 95% confidence (black axis). To compare with the fraction of G4/R-loop (or R-loop) from Fig. 6c (right *y*-axis with a blue circle and gray circle for NT-PQS and Control, respectively), the two graphs are overlaid. The exact data and *P*-values are provided

in the Source data file. ****P* < 0.0005, NS: nonsignificant (two-sided unpaired *t* test). **d** Promoter supercoiling density changes (left *y*-axis with the blue bar and gray bar for PQS and Control, respectively) with the fraction of G4/R-loop (or R-loop) (right *y*-axis with a blue circle and gray circle for PQS and Control, respectively) over transcription time were calculated by event-driven stochastic simulation. For details on the simulation, refer to Supplementary Fig. 13 and Supplementary document. **e** Summary schematic of PQS effect on transcription in supercoiled DNA, illustrating the interplay between G4/R-loop formation and transcription initiation events. The supercoiled DNA exhibits a 10-fold enhanced transcription rate due to pre-existing negative supercoiling. As transcription progresses, accumulated negative supercoiling induces transient R-loop formation, which in turn triggers G4 formation, stabilizing the R-loop structure and relaxing the DNA near the promoter. The stable formation of the G4/R-loop hinders RNAP loading due to superhelical relaxation, consequently slowing down the transcription rate.

interplay between supercoiling and transcription regulation. In conclusion, our investigation highlights the critical role of G4 and R-loop structures in regulating transcription initiation dynamics through modulation of DNA superhelicity at the promoter region. The cooperative formation of the G4/R-loop effectively suppresses the heightened transcriptional activity by impeding RNAP loading onto the promoter. Through FRET analysis and event-driven stochastic simulations, we elucidated the underlying mechanisms by which G4/R-loop structures alter DNA supercoiling and promoter accessibility, ultimately influencing transcription initiation rates (Fig. 7e).

## Discussion

Transcription imparts asymmetric topology on DNA by underwinding or negatively supercoiling the upstream DNA and overwinding or positively supercoiling the downstream DNA. While the underwound DNA becomes more accessible for the RNAP to engage with the promoter, it also enhances RNA transcript to anneal with the template DNA, forming R-loops. Furthermore, the R-loop can induce G4 formation on the non-template strand as both R-loop and G4 occur coincidently in guanine-rich sequences. Therefore, negative supercoiling facilitates both RNAP binding and R-loop/G4 formation[2,10,11]. Nevertheless, the structure-function relationship of R-loop/G4 formation and transcription in the context of topologically constrained DNA has not been revealed. Here, we employed ensemble transcription readout, EMSA, and single-molecule assays to systematically investigate how PQS impacts transcription in a topologically constrained DNA and elucidate the underlying mechanism.

### R-loop and G4 form synergistically in supercoiled DNA

Here, our innovative single-molecule plasmid constructs enabled us to understand the interplay between negative supercoiling, G4, and R-loop formation. We observed that frequent and short-lived R-loops stimulate subsequent G4 formation in the complementary NT- PQS strand in a supercoiled DNA. Notably, 97.5% of G4 is nucleated by the short-lived R-loops, which drives the formation of stable G4/R-loop (Fig. 5c and d). Previously, single-molecule R-loop footprinting (SMRF-Seq) was utilized to understand the mechanism of R-loop formation in plasmid constructs[61]. The study revealed that R-loops can form in regions close to the TSS, even in sequences unfavorable for R-loop formation, which was not observed in linear DNA. However, as transcription reaches more R-loop-favorable downstream sequences, the increase in negative superhelicity stabilizes R-loops' extension into the favorable regions. We directly observed the initial stage of R-loop formation occurring near TSS. Multiple rounds of short-lived R-loops represent that transcription-generated negative supercoiling is required to introduce a stable G4/R-loop. Contrary to the supercoiled DNA, linearized PQS only forms a long-lived R-loop and subsequent G4 formation, which infrequently occurred (3/138) with very slow kinetics (Supplementary Fig. 8). These results indicate that the short-lived R-loops may not be stable enough to form a

long-lasting structure. However, they provide sufficient time for G4 folding in the complement strand by enhancing strand separation in supercoiled DNA. The intrinsic negative supercoiling within the plasmid cannot explain why the short-lived R-loop nucleates G4. We hypothesize that G4 formation is facilitated by the additional negative supercoiling generated by the transcription activity and the concomitant transient R-loops. Furthermore, the presence of G4 in the complementary strand, in turn, dramatically enhances the strand separation for R-loop extension, as evidenced by the higher rates of R-loop formation in PQS than in the non-PQS constructs (Fig. 6e). In addition, the supercoiled PQS construct exhibited only short-lived R-loops that quickly transitioned to G4-stabilized R-loops, resulting in 100% of detected R-loops being associated with stable G4 formation. In contrast, the R-loops formed in the non-PQS construct were significantly less stable (Fig. 6c). These observations imply that the synergistically formed R-loop and G4 can effectively relax the highly constrained state of the negatively supercoiled DNA[59].

### DNA Supercoiling is the feedback mediator of transcriptional regulation, which is governed by G4/R-loop formation

We revealed how DNA supercoiling serves as an instant feedback mediator to regulate transcription in real-time. Pre-existing negative supercoiling enhances RNAP accessibility at the promoter, which induces a transcriptional burst (Fig. 5g). Subsequently, accumulated negative superhelicity drives G4/R-loop formation (Fig. 5h). Once the G4/R-loop forms, it absorbs negative superhelicity through alternative base pairing and strand twisting, allowing the rest of the domain to be partially or fully relaxed (Fig. 2). The topological release by G4/R-loop formation leads to subsequent repression in transcription (Figs. 1 and 7). In agreement with our work, Stolz et al. revealed that R-loops of lengths with 80 ~ 150 bp efficiently relaxed 3- to 4-kb negatively supercoiled plasmids by absorbing 15–20 supercoils[61]. Notably, although the sequence context of our PQS-NT or control in pUC19 plasmid is different from their R-loop-prone sequence, we have a similar size of R-loop with a comparable size of plasmid (3.7 kb) (Supplementary Fig. 13). These results corroborate to indicate that negative supercoiling is a primary driving force of R-loop extension, as previously expected[44,61,62]. Moreover, the propensity for R-loop formation is significantly increased by PQS-NT due to the subsequent G4, which stabilizes the R-loop (Fig. 6 and supplementary Fig. 3). The G4/R-loop also induces DNA topological relaxation (Fig. 2) and a significant attenuation of transcription output (Figs. 1 and 7).

Negative superhelicity represents a torsionally restrained high-energy state for DNA[61]. Stable R-loops serve as nonenzymatic means of relieving topological stress, as Stolz et al. described[61]. G4 formation in the NT strand facilitates and accelerates R-loop-mediated topological relaxation. Therefore, the G4/R-loop can function as a molecular switch that rapidly shuts down the transcription by absorbing negative supercoiling.

Furthermore, contrary to previous reports suggesting that R-loops impede RNAP movement[63,64] our experiments showed no evidence of truncated RNA (Supplementary Fig. 13). In addition, the dwell time of the high FRET state in the [FRET2] construct, indicative of the transition rate from RNAP initiation to the elongation complex, was not significantly affected by DNA superhelicity (Fig. 7b and Supplementary Fig. 12). These results shed new insights into understanding the role of R-loop dynamics in transcription regulation.

## Unraveling the dynamic complexity of transcription regulation by G4/R-loops

Our previous study showed that PQS-NT in linear DNA assists RNAP elongation, and G4/R-loops facilitate RNAP movement, increasing mRNA production[41]. Conversely, the same DNA sequence in topologically constrained circular DNA results in a burst of transcription followed by suppressed transcription (Fig. 1b, d). This contrasting impact indicates that the G4/R-loop formation can give rise to either transcription enhancement or repression depending on the context of the template DNA. For example, the transcriptional suppression by G4/R-loop mediated topological relaxation can be significantly altered when the gene has double-strand DNA breaks induced by a genotoxic reagent or endogenous DNA break.

In addition, this study identifies a position-dependent effect on transcriptional suppression influenced by R-loop formation dynamics. Recent studies (Belotserkovskii et al., 2022; Roy et al.) suggest that G-rich clusters within PQS regions near the transcription start site (TSS) facilitate the initiation of transcription-induced R-loop formation[37], with R-loops forming more readily when PQS is positioned close to the TSS[65]. When PQS is located further downstream, the increased RNA transcript length decreases the likelihood of R-loop initiation and stabilization[65], which may explain why transcriptional suppression is strongest near the TSS and alleviates as PQS moves downstream (Fig. 2). This biphasic pattern likely reflects the initial dynamics of R-loop formation, yet it warrants further investigation. Our results corroborate this, as we observed that PQS sequences close to the TSS (NT-sp10, 15, 20) exhibit the highest transcription suppression, with a gradual reduction as PQS is positioned further downstream (NT-sp25, 30) before the suppressive effect increases again at more distal positions (NT-sp35, 45, 60). This pattern underscores that the suppression effect is not linear but instead shows a biphasic dependence on PQS distance from the TSS, potentially linked to the dynamics of R-loop initiation.

We also demonstrated that the extent of transcription suppression depends on the DNA superhelicity, modulated by Top1 either pretreatment or treatment during transcription (Fig. 3). In addition, the position of PQS controls the transcription activity, i.e., even under the same level of negative supercoiling in DNA, the transcription level is impacted by the proximity of PQS to the TSS (Fig. 2d). Moreover, transcription activity relies on the mono-valent cation that alters G4 stability, which impacts the degree of topological relaxation (Supplementary Fig. 3). These observations demonstrate a fine-tuned regulatory mechanism mediated by the noncanonical DNA structures that operate under DNA supercoiling, sequence context, and buffer conditions. In addition, enzymes, including topoisomerases G4 helicases such as DHX36, RNase H, and Senetaxin, can dynamically change the DNA structure. Together, they modulate transcriptional activity. Therefore, we suggest that the transcriptional regulation via G4/R-loops is not merely a static or uniform process but a highly dynamic switching mechanism. This complexity warrants a deeper exploration of the molecular players governing transcriptional regulation, paving the way for more in-depth understanding and potential therapeutic targeting in conditions where these processes are dysregulated.

In summary, our study sheds light on the multifaceted nature of transcription regulation by G4/R-loops in varying positional, topological, and ionic conditions. It opens avenues for further research into the adaptive responses of transcriptional machinery under varying genomic contexts and stress conditions.

# Methods

## DNA preparation for beacon and gel retardation assays

To generate PQS-inserted plasmids with varying PQS positions relative to the transcription start site (TSS), we constructed a plasmid containing a GFP-expressing gene with the PQS sequence, c-MYC (5′-GGGTGGGTAGGG-3′), the T7 promoter, and the rrnB T1 terminator (Supplementary Table 1). For the sp30 construct, the transcription region spans 941 bp from the T7 promoter to the rrnB T1 terminator within the total 3682 bp plasmid. This region length varies depending on the PQS position. The rrnB T1 terminator was included to ensure proper transcription termination and prevent read-through by T7 RNA polymerase, as supported by previous studies[66]. Plasmids were cloned into the pUC57 vector using the NEBuilder Hifi DNA Assembly Kit (New England Biolabs) and amplified in *E. coli*, followed by purification with the OMEGA E.Z.N.A.® Plasmid DNA Mini Kit I (Omega Bio-tek). Linearized DNA templates were prepared by digesting the plasmids with EcoR I (New England Biolabs) and purifying with the QIAquick PCR Purification Kit (QIAGEN). To generate relaxed DNA templates with different degrees of superhelicity, 5 µg of plasmid was incubated with 10, 5, 3.3, and 2.5 units of *E.coli* DNA topoisomerases I (New England Biolabs) for 30 min at 37 °C, and followed by purification using QIAquick PCR Purification Kit (QIAGEN).

## Beacon assay

Ensemble in vitro transcription assay was performed by TECAN Spark plate reader at 37 °C. For each assay, 1 nM of each DNA template and 1.25 U/µL T7 RNAP (New England Biolabs) were mixed in 100 µL transcription buffer containing 40 mM Tris-HCl pH 8.5, 50 mM KCl, 6 mM Mg2Cl, 2 mM spermidine, 1 mM dithiothreitol and 2 U inorganic pyrophosphatase. Each sample was pre-incubated with a 400 nM molecular beacon probe (Supplementary Table 1) and loaded onto a 96-well transparent plate (Thermo Fisher Scientific). The reaction was initiated by adding 1 mM NTP mix (New England Biolabs). Cy3 was excited at $\lambda_{ex} = 545 \pm 10$ nm, and the fluorescence emission was collected per min at $\lambda_{em} = 570 \pm 10$ nm. The linear portions (first 20 min) of Cy3 intensity curves were fitted with a linear function for analysis (Supplementary Fig 17) as described in our previous work[41]. Consistent with our previous methodology, we excluded the initial ~400 s of data from the analysis to allow for temperature stabilization. This ensured consistency across all measurements.

To test the effect of Topo I (New England Biolabs) during transcription, the measurements were carried out in the presence of 0, 0.015, and 0.15 U Topo I.

## Gel retardation assay

Transcription using T7 RNAP (1.25 U/µL) was carried out with 100 ng of each DNA template in 20 µL transcription buffer at 37 °C for 1 hr. The reaction was stopped with an addition of 1 µM 22-mer T7 promoter DNA and treated with 0.01 mg/ml RNase A (Thermo Fisher Scientific) for 30 min at 37 °C to degrade excess free RNA. To address R-loop-mediated topological shift, the transcribed sample was treated with 0.125 U/µL RNase H (New England Biolabs) and 0.01 mg/ml RNase A (Thermo Fisher Scientific) for 30 min at 37 °C. All samples were purified using the QIAquick PCR Purification Kit (QIAGEN). Half of the purified samples (10 µL in 20 µL) were mixed with 2 µL of 50% glycerol and separated by 1% agarose gel electrophoresis. The gel was run at 100 V for 70 min and post-stained with SYBR Green II. The image was taken by gel imager (Amersham imager 600) with $\lambda_{ex} = 520$ nm. To quantify the shift from supercoiled to relaxed states, as shown in Fig. 2d and e, we obtained gel band intensities from the gel images using Fiji. After importing the gel images into Fiji, we measured the intensity of each band to determine the distribution of gel bands. For

further analysis of topological shifts, we used OriginPro 2018 to fit a Gaussian function to the supercoiled state. The area under the Gaussian curve represented the intensity of the supercoiled bands. We then calculated the relaxed fraction as follows:

$$Relaxed(\%) = \left( \frac{Total - supercoiled}{Total} \right) \times 100$$

where "Total" represents the total area in the gel band distribution, and "Supercoiled" refers to the area under the Gaussian curve fit for the supercoiled bands (Supplementary Fig. 16).

### DNA preparation for Single-molecule PIFE and FRET assay
The DNA oligonucleotides (Supplementary Table 1) were purchased from IDT with an internal amine or biotin modification. The internal amine was used for Cy3 (oligo1) and Cy5 (oligo2) labeling. The biotin modification (oligo3) was used for surface immobilization. The amine-modified oligomers were dissolved in $H_2O$ to make 500 μM stock. DNA (40 μL, final concentration 200 μM) was mixed with 0.2 mg Sulfo-Cyanine3 or Sulfo-Cyanine5 NHS-ester (Lumiprobe), 50 μL 1 M sodium bicarbonate buffer, and 10 μL H2O. The reaction was kept at 40 °C for 2 hr in a shaking incubator. The excess dye was removed by ethanol precipitation and followed by HPLC purification (Agilent Technologies)[67].

To synthesize relaxed and supercoiled DNA constructs[57,68], 10 μg of plasmid was digested by 35 U of Nt.BbvCI in 210 μL of 1 × NEBuffer™ 2.1 (New England Biolabs). After the digestion, 146 pmol (~ 30-times molar excess) of phosphorylated oligo 1, 2, and 3 were added into the reaction mixture for [FRET1]. The mixtures were heated at 90 °C for 5 min and slowly cooled to room temperature (1 °C per min). To generate a relaxed DNA construct, 400 U of T4 DNA ligase and 5 U of T4 DNA polymerase were added into the reaction mixtures in the presence of 10 mM of DTT and 2 mM of ATP (final concentrations). To generate supercoiled DNA construct[68], 5 μM ethidium bromide (EtBr, final concentration) was added in the mixture. The mixtures were incubated at 37 °C for 90 min to seal the nick and followed by 50 U of T5 exonuclease treatment at 37 °C for 60 min. All samples were purified using the QIAquick PCR Purification Kit (QIAGEN). For the EtBr-treated sample, it further needed to purify using Butanol extraction before the final clean-up process[67]. We typically obtained the product with ~ 60% yield. To generate [FRET2], phosphorylated oligo 1 and oligo 3 were inserted to non-template strand, and then oligo 2 was inserted to the other strand. The final yield was ~ 30%.

### Single-molecule FRET and PIFE measurements
All the single-molecule experiments were carried out by using a home-built prism-type total internal reflection fluorescence microscope at room temperature (23.0 ± 1.0 °C). Fluorescently labeled long DNA was diluted to 1 nM with 20 μL of 10 mM pH 8.5 Tris-HCl and 50 mM NaCl (T50) and immobilized on a PEG-coated quartz slide pretreated with neutravidin (0.05 mg/ml). The imaging buffer was prepared freshly by mixing transcription buffer with an oxygen scavenging system (1 mg/ml glucose oxidase, 0.8% v/v glucose, ~ 2 mM Trolox, and 0.03 mg/ml catalase). 532 nm and 641 nm solid-state diode lasers were employed to excite Cy3 and Cy5. Fluorescent emissions were separated by a dichroic mirror (cutoff: 640 nm) and collected by electron-multiplying CCD (Andor). All data was recorded with a 100 ms time resolution by smCamera software and analyzed with Interactive Data Language. Single-molecule traces and histograms are further analyzed with scripts written in MATLAB. IDL and Matlab scripts were provided by Dr.Taekjip Ha's lab. The smFRET histograms were normalized and fitted to Gaussian distributions using Origin 2018.

### Statistical analysis for smFRET
For each FRET histogram presented in Figs. 5f, h, and 6e, FRET efficiency values were compiled from ~ 6000 molecules, obtained from 15 ~ 20 movies. Signals containing only the donor (Cy3) were excluded. The data were then analyzed using a Gaussian distribution function to generate the histograms. For the quantification of the transcription frequency in Fig. 5g, we quantified the number of PIFE peaks before it reaches high FRET state. Most of supercoiled NT-PQS molecules exhibited a high FRET state (E ~ 0.9) in 200 s, it must convert the dwell time of low to mid FRET state into 200 s with the number of PIFE peaks.

### Reporting summary
Further information on research design is available in the Nature Portfolio Reporting Summary linked to this article.

## Data availability
Source data are provided in this paper.

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

## Acknowledgements

We thank all the members of the Sua Myong and Taekjip Ha laboratories for their helpful comments and constructive criticisms. This work was supported by NIH R01 GM149729-04.

## Author contributions

J.H., C.Y.L., and S.M. designed experiments. J.H., T. P., H. L., and A.C., conducted experiments. S.B. and S.T. performed the computational studies. R. Small supervised the work. J.H., C.Y.L., and S.M. wrote the paper.

## Competing interests

The authors declare no competing interests.
