## [Transparent Peer Review file · Nature Communications]

DNA supercoiling-mediated G4/R-loop formation tunes transcription by controlling the access of RNA polymerase

Corresponding Author: Dr Sua Myong

Version 0:

Reviewer comments:

Reviewer #1

(Remarks to the Author)

The authors have performed bulk and single molecule investigations comparing the impact of potentially G-quadruplex forming sequences (PQS) on transcription for linearized or supercoiled (in plasmid form) DNA. The study primarily focused on the case when PQS was in the non-template strand and in all cases the PQS were placed downstream of the transcription start site (TSS). The authors demonstrate that PQS differently impact transcription in linearized DNA compared to supercoiled DNA and propose a model to describe the variations. The article is well-written with an adequate level of detail. The authors have used similar techniques in earlier studies and are experienced on the topic. The conclusions of the article are largely in-line with the literature on relevant topics; however, the combination of single molecule and bulk assays provides access to a higher level of detail.

Major Comments:

PQS are concentrated in promoters and regulatory sites and coding sequences are poor in PQS. Yet, the study presents PQS downstream of transcription start site as mechanisms for transcription regulation. Is there any evidence for this mechanism taking place or playing significant role in a physiological setting?

Why did the study focus on PQS in non-template strand? Except for two measurements in Fig. 1B-C, all other data is acquired for the condition PQS in NT. The authors should mention this in various part of the manuscript as most of their conclusions are made generally, which gives the impression that the same results might be expected (or were obtained) for the case of PQS in T.

When determining the transcription rate, how do the authors determine the linear part? Depending on the cutoff, the slope of the fit line will change. Did the authors use a consistent time range for the part to be fitted with a line? Is there a justification for selecting a given range as linear?

My specific comments, which are minor in nature, are listed below (first as they appear in text followed by figure specific comments):

Page 4, Line 111: Section title reads: "PQS promotes transcription in linear DNA but suppresses it in supercoiled DNA". The control and PQS-NT appear to have the same rate in supercoiled DNA (Fig. 1B-C). Later in Fig. 1e, the authors show that sp30 is an exceptional case where the transcription of PQS-NT is similar to that of the control (unlike all other constructs). So, I think a construct different from sp30 should be presented in Fig. 1b to show suppression for supercoiled DNA.

Page 4, Line 117: How do the authors ensure the beacon binds only to the transcribed RNA and not the non-template DNA strand when a transcription bubble is generated?

Page 5, Line 141: The authors wrote: "the PQS located proximal to TSS (NS-sp10, 15, 20) displayed the highest level of transcription suppression, which is gradually alleviated downstream (NT-sp25, 30) and again suppressed further downstream (NT-sp35, 45, 60)." Any idea why such a pattern is observed?

Page 5, Line 145: "Taken together, 144 PQS in supercoiled DNA suppresses transcription position-dependent,..." needs fixing.

Page 6, Line 192: "Our results indicate that the G4/R-loop structures inhibit transcription only in the context of negatively supercoiled DNA." The authors have not demonstrated the validity of this statement when PQG is in the template strand. Almost all measurements have been for PQS-NT case. Given this, how do they justify this statement?

Page 7, Line 216: Have the authors confirmed whether EtBr does not impact the transcription beyond inducing negative supercoiling? It presumably changes the stability and local helical structure of the DNA, which would impact transcription and associated processes. Also, how does the level of supercoiling (including fraction of molecules that undergo supercoiling and extend of supercoiling) achieved with EtBr compare with that achieved with 5 μ M Ethylenediaminetetraacetic acid?

Page 8, Line 245: The authors wrote: "The FRET histogram analysis revealed that the high FRET peak diminished in G4 destabilizing conditions (Figure 5f, no M+), confirming the long-lived high-FRET as a stably folded G4 state (Figure 5f)." Do the authors expect a G4 to fold under these conditions since the majority of high FRET population appears to remain under these conditions? Most G4 structures would not fold under these conditions.

Page 8, Line 257: "Based on the result, the 257 stable high FRET state likely encompasses both R-loop and G4 (G4/R-loop)." This statement appears to contradict the earlier assignment in which the high FRET state was exclusively attributed to G4. The implications of the observed effects significantly change depending on which interpretation is assumed. Can the authors clarify?

Pages 8-9: Most of the statements about Figure 5 are based on sample traces and have not been quantified in a way that justifies the conclusions. Such statements are often weak as they demonstrate significant variations among different traces. The authors should consider performing more quantitative analysis for this section.

Page 10, Line 328: "In conclusion, our investigation highlights the critical role of G4 and R-loop structures in regulating transcription initiation dynamics through modulation of DNA superhelicity at the promoter region." As all constructs used in this study had PQS downstream of TSS, I am not sure whether their effects could still be considered as "transcription initiation".

Fig. 1b-c: How do the control measurement compare in linear and supercoiled configurations without normalization?

Fig.1. Dashed lines in this figure appear solid on my screen for this figure.

Fig. 1e: Error bars are quite large. Are the differences significant? Have the authors performed any statistical test to establish the significance of the differences?

Figure 2c: The transcription rate for NT-sp30 in Fig. 1e is essentially same as the control (within the experimental error bars). However, substantial shifts are observed for this construct after transcription in Fig. 2c. This would suggest those shifts are not the cause of the transcription suppression. Given this, can the authors justify the following statement in Page 5, Line 161: "Based on the higher suppression seen in NT-sp10, the shifted bands likely correspond to DNA conformations that suppress transcription." Does the shift pattern in the control and that of sp30 look similar?

Figure 2d: How did the authors quantify the shift to relaxed state?

Figure 3a-b: The linear part of the curves (until about 20 min) appear to overlap, which would have suggested the same rate; however, the rates in (b) are not the same. Is there a criterion used to determine which part of the curve is treated as linear?

Figure 3b: Was the control construct supercoiled in Fig. 3b? If yes, shouldn't topoisomerase also resolve negative supercoiling in this construct and give rise to enhanced transcription rate?

Figure 5f: How many molecules were included in the histogram in Fig. 5f? The differences between first and second panels in Fig. 5f appears negligible.

Figure 5 caption: The authors wrote: "The expected E of R-loop and G4 is ~ 0.7 and ~ 0.9 , respectively." How has this been established? Also, why is the entire range between 0.3-0.8 marked as R-loop if the expected range for it is around $E \sim 0.7$?

Figure 5d caption: The authors wrote: "d) A closer examination of the light gray area of (c) reveals a distinct temporal shift between PIFE and FRET peaks." First, I do not see a light gray area in (c). Also, have the authors performed any analysis to justify this statement (other than observing it in a particular trace)? The time difference between the two events is very small. In fact, the onset of the events appears the same within time resolution of the experiment, but the donor emission appears to reach its maximum value before the FRET change reaches its maximum.

Figure 6 caption. I assume (c) is for -sp30 and (d) is for control. The authors should clarify.

Figure 7a caption: The authors wrote: "...featuring Cy3 and Cy5 positioned 4 bp (-4) and 19 bp (+19) downstream from the TSS, respectively." Shouldn't "(-4)" be "(+4)"?

Reviewer #2

(Remarks to the Author)

In this study, Jihee Hwang, et.al used single-molecule fluorescence and other techniques to study the interrelationships among RNA polymerase, promoter, G4, R-loop, and DNA supercoiling, and found that the formation of G4/R-loop changes the helical state of promoter DNA, thereby affecting the binding ability of RNA polymerase. This study reveals a possible way in which gene transcription is affected by G4/R-loop, and provides very meaningful information for our understanding of the function of PQS downstream of gene promoters. However, there is still room for improvement in some details, as described below.

1. In the transcription experiment of supercoiled plasmid, we did not see the authors describe how long the T7 RNAP transcription region is. If there is no terminator, T7 RNAP may transcribe the entire plasmid DNA, resulting in a global change in the helical state of the plasmid. Should the authors limit the length of T7 transcription to the region related to transcription initiation, so as to study the effect of R-loop formation in the promoter region on the change of the local helical state and the binding ability of T7 RNAP?
2. In line 111, the author claimed that "PQS promotes transcription in linear DNA but suppresses it in supercoiled DNA". however, the results in Figure 1b and 1c did not show PQS suppresses transcription in supercoiled plasmid.
3. There is no PQS sequence labeled in any of the result figures. What type of G-quadruplex can it form? Are there multiple conformations?
4. PQS has a more obvious inhibition on transcription only when it is within 20 bases of the T7 promoter (Figure 1e). Is it possible that the formation of G4 and R-loop in the aborted transcription region causes the T7 promoter to partially unwind and thus cannot be bound by T7 RNAP?
5. In Supplementary Figure 13, the amount and size of R-loops in NT-sp10 and NT-sp30 are similar, but in Figure 2b, the proportion of relaxed states in NT-sp10 is higher than that in NT-sp30. How can this be explained?
6. Lines 321-323, Supplementary Figure 12 were conducted under G4 destabilizing condition. How do the authors conclude that "The level of negative supercoiling is likely reduced upon the formation of G4/R-loop structures to a level similar to that of a relaxed DNA".
7. Figure 5g shows that the transcription frequency of supercoiled plasmids is 10 times higher than that of linear and relaxed plasmids. However, in the Beacon Assays in Figures 1 and 3, the transcription efficiencies of different types of plasmids are not that different. It seems that the results of different experiments are inconsistent?
8. The FRET1 model in Figure 5a may require further confirmation. As described in a previous study (Liu JQ, Xiao S, Hao YH, Tan Z. Strand-Biased Formation of G-Quadruplexes in DNA Duplexes Transcribed with T7 RNA Polymerase. *Angew Chem Int Ed Engl.* 2015 Jul 27;54(31):8992-6.), when PQS is located on the non-template DNA strand, G-quadruplex formation begins when the 3' end of RNA synthesis is approximately 7 nt away from PQS.
9. As mentioned in question 3, PQS may form G-quadruplexes with different conformations. Is it possible that the phenomenon of mid FERT and high FRET is caused by the difference in G-quadruplex conformation?
10. In Figure 5, the description of no M+ is inaccurate. In fact, each NTP in the transcription reaction usually brings 2-3 monovalent cations, and the storage buffer of T7 RNA polymerase also brings a certain concentration of monovalent cations.

Reviewer #3

(Remarks to the Author)

Manuscript by Hwang et al applied ensemble fluorescence, single molecule FRET-PIFE and gel-based assay, reports the cross-regulation effect of transcription, negative supercoiling and G4/R-loop formation. Detailed understanding of these events is crucial for the proper contextual placement of in vitro transcription experiments within the general framework of gene expression in cellular milieu.

Specific comments:

- 1). In the first part of the results, authors used "linear" DNA as control compared to "supercoiled" circular DNA to conclude the DNA topology affects transcription. Would the authors provide a set of measurement use "nicked" circular DNA to interpretate this effect? With the nicked site more downstream of the transcription which can release the supercoiling generated by transcription while maintain the circular form of the DNA.
- 2). Would author provide the detail of the "Beacon" specific binding place in the overall sequence, is it near the TSS or more downstream of the transcription sequence? Because DNA topology also influence the transcription processivity.
- 3). In line 126 that "we obtained ~30%-40% enhanced mRNA production rate in PQS-NT compared to control and PQS-T, respectively", the authors would say "~30% and ~40%" or say "compared to both control and PQS-T".
- 4). Would the authors be more specific on how to calculate the "transcription rate (%)" in Figure 1"? By looking at the Figure 1, the increase in of fluorescence intensity at initial time and the final saturation plateau are both different, given that not only the transcription speed and the final mRNA amount are both different.
- 5). Line 141, I assumed the authors want to say "downstream of TSS in supercoiled plasmid" because the linear DNA showed no difference in transcription rate.
- 6). Line 153, the authors choose NT-sp10 and NT-sp30 based on high and low level of suppression, would the author do one more test on mid-level of suppression such as NT-sp50 or NT-sp60? Because based on figure 1e, the suppression of the PQS to TSS position is not simple increase or decrease, but two phase. Do the authors have any hypothesis why this happen?
- 7). Would the authors explain how they define/calculate the "shift to relax state" in figure 2?

- 8). In line 175-176, the authors said “highest to lowest R-loops are detected for NT-sp10, -sp15, -sp20, -sp30, -sp35 (Figure 2d) which is inversely related to the transcription rate (Figure 1e)”. It is correct except -sp30 and -sp35 since there is a decrease trend in Figure 1e.
- 9). In supplementary Fig2b, how did the authors normalize the distribution?
- 10). Line 198, authors said that “pre-treating DNA with Topo 1 showed a dose-dependent increase in transcription in the NT-sp10 and NT-sp30”. It shouldn’t be described as dose dependent since it’s a factor of efficiency, if treat sufficient time all supercoiling will be released.
Also, the authors should be how they quench the Topo I reaction.
- 11). Would author specify the Topo I concentration in certain treat time in the figure 3 instead of highest-high-mid-low-no?
- 12). Line 216-217, would the authors mark the lane with number in figure 4b left, otherwise the lane 1 in the figure is a ladder.
- 13). Fig 4b, the lane of relaxed DNA seems still supercoiled, but with less superhelicity.
- 14). In the [FRET1] strategy, the author indicate that the mid-FRET state is $E \sim 0.7$, in figure 5f and e, the R-loop state is (blue box) is span from $E \sim 0.4$ to $E \sim 0.8$ stage made it confusing. By looking at the traces in supplementary fig7, it is easy to understand the transition is a procedure and it progress fast and made the FRET efficiency looks spanning a wide range. I suggest the author to better clarify in the text.
- 15). Line 289, will it be the Figure 6c, d, bottom rather than the top?
- 16). Line 317, “ie., 200 sec, 10 and 30 minutes” however the figure 7c marked as “0, 10 and 30 mins”.
- 17). Would the author explain better how to get the supercoiling density in figure 7d, rather than just cite the paper?
- 18). The supplementary information missing supplementary table 1.
- 19). Line 363, “linearized PQS showed only a long-lived R-loop and subsequent G4 formation, which infrequently occurred (3/138) with very slow kinetics.” Here the authors want to describe the “linearized PQS only has the scenario of long-lived R-loop” or “the linearized PQS also has short-lived R-loop, but only the long-lived R-loop form G4”?
- 20). Line 374, the authors said supercoiled PQS construct form 100% G4/R-loop structure, but according to fig 6e, only 60% formed G4/R-loop. Would the authors clarify this?
- 21). Line 496, did the authors want to say “EtBr” rather than EDTA?
- 22). Line 509, the buffer recipe in the bracket looks like “imaging buffer”, the authors might mark it after the “transcription buffer” by mistake.
- 23). Would the authors plot the Fig 2f and Fig 3e in a quantitative way?
- 24). In the legend of figure 5(d) state “the light grey area of (c)”, but there is no grey area in the (c). And the color box “blue box” and “red box” “grey box” in the figure is not easy distinct, would the authors edit it better?

Version 1:

Reviewer comments:

Reviewer #1

(Remarks to the Author)

The authors have made a commendable effort to respond to my comments and make the necessary modifications to the manuscript. I do not have further comments about the revised manuscript.

Reviewer #2

(Remarks to the Author)

The authors have carefully responded to my questions and made appropriate revisions. I support the publication of the revised paper.

Reviewer #3

(Remarks to the Author)

The authors successfully addressed my comments in the responses and I therefore support

the publication of this manuscript.

POINT BY POINT RESPONSE TO REVIEWERS' COMMENTS

(Our responses are headed by left arrows in navy blue)

We are incredibly grateful for the reviewers' insightful criticisms and comments. We spent the last three months performing new experiments, reproducing and confirming previous results, and reanalyzing data according to the reviewers' detailed suggestions. As a result, our results have been greatly strengthened, as seen in the revised manuscript. We appreciate the time and effort reviewers put into reviewing and commenting on our manuscript, and we hope we have addressed all the concerns.

Reviewer #1 (Remarks to the Author):

The authors have performed bulk and single molecule investigations comparing the impact of potentially G-quadruplex forming sequences (PQS) on transcription for linearized or supercoiled (in plasmid form) DNA. The study primarily focused on the case when PQS was in the non-template strand and in all cases the PQS were placed downstream of the transcription start site (TSS). The authors demonstrate that PQS differently impact transcription in linearized DNA compared to supercoiled DNA and propose a model to describe the variations. The article is well-written with an adequate level of detail. The authors have used similar techniques in earlier studies and are experienced on the topic. The conclusions of the article are largely in-line with the literature on relevant topics; however, the use of single molecule and bulk assays provides access to a higher level of detail.

Major Comments:

PQS are concentrated in promoters and regulatory sites and coding sequences are poor in PQS. Yet, the study presents PQS downstream of transcription start site as mechanisms for transcription regulation. Is there any evidence for this mechanism taking place or playing significant role in a physiological setting?

- ⇒ As the reviewer points out, PQS is located in promoter region i.e transcription start site (TSS). Our PQS is in **5' untranslated region (5' UTR)**, where PQSs are mostly densely populated. The 5' UTR is widely recognized for its role in transcription and translation regulation (Halder K. et al., *Nucleic Acids Res.* (2009); Holder I. T. et al., *Chem. Biol.* 21 (2014); Agarwal T. et al., *Biochemistry* (2014))
- ⇒ As noted in our manuscript, Verma A. et al. (Reference #18) identified 38,757 PG4 sequences within ± 1 kb of the TSS across 25,706 annotated human genes, with 8490 (21.9%) residing in the 5' UTR, a region known to play critical regulatory roles. Given this, we agree that specifying "5'-UTR" rather than "downstream of the TSS" better highlights the physiological relevance of PQS in transcription regulation.
- ⇒ To clarify further, we have revised the manuscript to state:

"line60: ~370,000 potential quadruplex-forming sequences (PQS) have been predicted in the human genome and are highly enriched in the promoter region of regulatory genes, splice junctions, and the 5' untranslated region (5' UTR) of transcriptionally active genes, particularly in oncogene promoters.¹⁴⁻¹⁸ Therefore, they are proposed to regulate gene expression and are targeted by small molecules for therapeutic intervention."

⇒ Additionally, references 14 to 18 in our manuscript support the potential regulatory impact of PQS in the 5' UTR. To further highlight the physiological evidence of PQS's role in the 5' UTR *in vivo*, we have included additional references (References #23–24), underscoring the importance of PQS in transcriptional control.

Why did the study focus on PQS in non-template strand? Except for two measurements in Fig. 1B-C, all other data is acquired for the condition PQS in NT. The authors should mention this in various part of the manuscript as most of their conclusions are made generally, which gives the impression that the same results might be expected (or were obtained) for the case of PQS in T.

⇒ In our previous work (Lee et al. 2020), we demonstrated that G4 formation in the template blocked the formation of R-loop and inhibited the transcription likely due to the structural obstacle to RNA polymerase. This leads to a negative outcome instead of a regulatory mechanism. Our data (Figure 1c) also indicated the reduction effect of template PQS was not related to supercoiling. Hence, this study mainly focuses on the non-template orientation and its correlation with supercoiling effect. To reflect this, we have included statements in the manuscript such as:

"Our previous work found that co-transcriptionally formed R-loops induce G4 formation in the non-template strand, which enhances transcription yield in linear DNA.

"For linear DNA, in agreement with our previous findings, we observed a ~30-40% increase in mRNA production rate in PQS-NT compared to both the control and PQS-T conditions (Figure 1B and 1C)."

"Our previous study showed that PQS-NT in linear DNA assists RNAP elongation, with G4/R-loops facilitating RNAP movement, thereby increasing mRNA production."

When determining the transcription rate, how do the authors determine the linear part? Depending on the cutoff, the slope of the fit line will change. Did the authors use a consistent time range for the part to be fitted with a line? Is there a justification for selecting a given range as linear?

⇒ We use a set time range for fitting the linear part across each set of measurements, following the same way used in our previous works (Lee et al., 2020 and 2023). As done previously, we excluded the initial few minutes (~400 seconds) from the analysis to account for temperature stabilization.

⇒ For clarification, we revised as follows:

"Line489: The linear portions (first 20 min) of Cy3 intensity curves were fitted with a linear function for analysis (Supplementary Fig 17) as described in our previous work41. Consistent with our previous methodology, we excluded the initial ~400 seconds of data from the analysis to allow for temperature stabilization. This ensured consistency across all measurements."

My specific comments, which are minor in nature, are listed below (first as they appear in text followed by figure specific comments):

Page 4, Line 111: Section title reads: "PQS promotes transcription in linear DNA but suppresses it in supercoiled DNA". The control and PQS-NT appear to have the same rate in supercoiled DNA (Fig. 1B-C). Later in Fig. 1e, the authors show that sp30 is an exceptional case where the

transcription of PQS-NT is similar to that of the control (unlike all other constructs). So, I think a construct different from sp30 should be presented in Fig. 1b to show suppression for supercoiled DNA.

⇒ We appreciate your valuable feedback. Our primary comparison is the transcription rates of PQS between **supercoiled** and **linear** DNA; our data consistently show that supercoiled constructs exhibit suppressed transcription rates compared to their linear counterparts with the same sequence. This difference highlights the regulatory effect of supercoiling, as illustrated by the combined data in Figure 1d.

⇒

⇒ To address your concern and provide clarity, we have updated the section title to:
"line 111: Distinct Transcriptional Regulation by PQS in the 5' UTR in Supercoiled and Linear DNA"

Page 4, Line 117: How do the authors ensure the beacon binds only to the transcribed RNA and not the non-template DNA strand when a transcription bubble is generated?

⇒ This is the same beacon we used in our previous studies (Lee C.-Y. et al., *Nat Commun*, 2020 and Lee C.-Y et al., *Nat Commun*, 2023), where we addressed the same concern. To confirm the specificity of the beacon binding to the transcript, we conducted both single-molecule beacon assays and gel analysis (Figure 1). These experiments demonstrated that the beacon predominantly binds to RNA rather than to the non-template DNA strand. We have included a reference to this method in the manuscript to clarify this point.

- ⇒ Additionally, we performed a control experiment in which the transcription reaction for NT-sp30 was conducted without the rNTP mix. Under these conditions, no signal was observed from the DNA template, indicating that the beacon signal arises solely from its binding to the transcribed RNA.
- ⇒ In conclusion, these results confirm the specificity of the beacon for the RNA transcript during transcription.

Page 5, Line 141: The authors wrote: "the PQS located proximal to TSS (NS-sp10, 15, 20) displayed the highest level of transcription suppression, which is gradually alleviated downstream (NT-sp25, 30) and again suppressed further downstream (NT-sp35, 45, 60)." Any idea why such a pattern is observed?

- ⇒ The observed position-dependent transcriptional suppression is likely influenced by the dynamics of R-loop formation. Previous work by Roy D. et al. (*Mol Cell Biol*, 2009) suggests that G-clusters play a key role in initiating transcription-induced R-loop formation. In our study, we used the c-MYC sequence, which contains a high density of G-clusters in the non-template DNA, making it particularly prone to R-loop initiation near the PQS. Additionally, a recent study by Dr. Hanawalt's group (Belotserkovskii, B. P. et al., *Biophys J*, 2022) provides further insight into R-loop formation mechanics. They demonstrated that R-loops occur more easily when the RNA transcript is shorter, which is the case when the PQS is located closer to the transcription start site (TSS). As the RNA lengthens with increasing distance from the TSS, it becomes more challenging for the RNA to pass through the DNA strands, reducing the likelihood of R-loop formation. This model may explain why transcriptional suppression is most pronounced when the PQS is proximal to the TSS and lessens further downstream. In our experiment (Figure 2), we observed that R-loop formation causes topological relaxation, leading to transcriptional suppression. However, the exact mechanism remains unclear and requires additional investigation to fully understand these dynamics.
- ⇒ We have incorporated a new discussion section to address this point:

"line430: Additionally, this study identifies a position-dependent effect on transcriptional suppression influenced by R-loop formation dynamics. Recent studies (Belotserkovskii et al., 2022; Roy et al.) suggest that G-rich clusters within PQS regions near the transcription start site (TSS) facilitate the initiation of transcription-induced R-loop formation³⁷, with R-loops forming more readily when PQS is positioned close to the TSS⁵³. When PQS is located further

downstream, the increased RNA transcript length decreases the likelihood of R-loop initiation and stabilization⁶⁵, which may explain why transcriptional suppression is strongest near the TSS and alleviates as PQS moves downstream (Figure 2). This biphasic pattern likely reflects the initial dynamics of R-loop formation, yet it warrants further investigation. Our results corroborate this, as we observed that PQS sequences close to the TSS (NT-sp10, 15, 20) exhibit the highest transcription suppression, with a gradual reduction as PQS is positioned further downstream (NT-sp25, 30) before the suppressive effect increases again at more distal positions (NT-sp35, 45, 60). This pattern underscores that the suppression effect is not linear but instead shows a biphasic dependence on PQS distance from the TSS, potentially linked to the dynamics of R-loop initiation."

Page 5, Line 145: "Taken together, PQS in supercoiled DNA suppresses transcription position-dependent,..." needs fixing.

⇒ The sentence is revised as below:

"line144: Taken together, PQS in supercoiled DNA suppresses transcription in a position-dependent manner, while the same PQS in linear DNA enhances transcription."

Page 6, Line 192: "Our results indicate that the G4/R-loop structures inhibit transcription only in the context of negatively supercoiled DNA." The authors have not demonstrated the validity of this statement when PQS is in the template strand. Almost all measurements have been for PQS-NT case. Given this, how do they justify this statement?

⇒ As we addressed in question two above, the inhibitory effect of PQS-template is unlikely to be the same as PQS-non template. We have revised the sentences as follow:

"line192: Our results suggest that G4/R-loop structures inhibit transcription primarily when PQS is located near the TSS in the non-template strand of negatively supercoiled DNA."

Page 7, Line 216: Have the authors confirmed whether EtBr does not impact the transcription beyond inducing negative supercoiling? It presumably changes the stability and local helical structure of the DNA, which would impact transcription and associated processes. Also, how does the level of supercoiling (including fraction of molecules that undergo supercoiling and extend of supercoiling) achieved with EtBr compare with that achieved with 5 μ M Ethylenediaminetetraacetic acid?

⇒ We thank you for highlighting this critical point. We fully recognize the importance of confirming that EtBr does not interfere with transcription beyond its role in inducing supercoiling. The reference study on plasmid modification (Wang, Y. et al., *ACS Omega*, 2019) used EtBr to introduce supercoiling and demonstrated that the resulting supercoiling density is highly dependent on the concentration of EtBr applied. In that study, phenol extraction was used to remove EtBr from the ligated DNA. In our work, we employed butanol extraction rather than phenol extraction, as it provides efficient and rapid purification for fluorescently labeled DNA oligonucleotides without interfering with downstream activity (Hwang, J. et al., *BioTechniques*, 2013). After butanol extraction, we conducted an additional step with column-based purification to ensure thorough removal of EtBr and ligation byproducts.

⇒ To further illustrate the impact of EtBr presence and removal on supercoiling density, we provided Supplementary Figs. 16 and 17.

a

b

c

Supplementary Fig. 16: EtBr remains on ligated DNA without butanol extraction.

(a) In the absence of butanol extraction (second lane of the gel, supercoiled -Butanol ext), ligated DNA containing EtBr migrates higher on the agarose gel compared to non-treated plasmid. However, following butanol extraction, the supercoiled reconstituted construct migrates to the same position as the untreated plasmid, as shown in Figure 4b. (b) Single-molecule imaging reveals residual EtBr on the plasmid DNA, observed as a red signal upon 532 nm laser excitation, due to EtBr's broad emission spectrum. (c) After butanol extraction, these red signals disappear, confirming the effective removal of EtBr and resulting in the

intended

low

a

FRET

signal

b

Supplementary Fig. 17: supercoiling density depends on EtBr concentration.

DNA treated with 5 μM EtBr (third lane) during the ligation process exhibits a supercoiling density similar to that of the non-modified plasmid. In contrast, DNA treated with a lower concentration of EtBr (0.5 μM) shows reduced supercoiling density, resulting in a more relaxed DNA structure. This confirms that supercoiling density is directly dependent on the concentration of EtBr as shown in Wang, Y. et al., ACS Omega (2019)³.

- ⇒ In Supplementary Fig. 16, we demonstrate the effectiveness of butanol extraction in removing residual EtBr from ligated DNA. (a) shows that when EtBr remains on the ligated DNA, it migrates higher than non-treated DNA on the agarose gel, suggesting an altered structure due to intercalation. (b) Single-molecule imaging reveals the presence of residual EtBr on the DNA, which emits a red signal upon 532 nm laser excitation, attributable to EtBr's broad emission spectrum. After butanol extraction (c), this red signal is no longer detectable even with the highly sensitive EMCCD, which is capable of single-molecule detection. This confirms the successful removal of EtBr and yields the intended low FRET signal.
- ⇒ In Supplementary Fig. 17, we examine the relationship between EtBr concentration and supercoiling density. DNA treated with 5 μM EtBr during ligation shows a supercoiling density

comparable to the non-modified plasmid. Conversely, when the EtBr concentration is reduced to 0.5 μM , we observe reduced supercoiling density, resulting in a relaxed DNA structure. Therefore, we confirmed 5 μM EtBr is suitable to generate negative supercoiled construct.

Page 8, Line 245: The authors wrote: "The FRET histogram analysis revealed that the high FRET peak diminished in G4 destabilizing conditions (Figure 5f, no M+), confirming the long-lived high-FRET as a stably folded G4 state (Figure 5f)." Do the authors expect a G4 to fold under these conditions since the majority of high FRET population appears to remain under these conditions? Most G4 structures would not fold under these conditions.

⇒ We thank you for raising this important point. In our previous work (Reference #41, Lee C.-Y. et al., *Nat Commun*, 2020), we confirmed that G4 formation does not occur in linear DNA under conditions lacking monovalent cations. However, in supercoiled DNA, we observed G4 structures forming even under G4-destabilizing conditions. This suggests that the structural constraints imposed by supercoiling may facilitate G4 formation, even in environments that would otherwise be unfavorable.

⇒ **Supporting evidence from prior studies:**

1. Liu *et al.* demonstrated that transcription acts as a driving force for G4 formation, with G4 structures forming downstream of the transcription start site (TSS) as a PQS is approached by a transcription bubble (Liu, J. *et al.*, *Angewandte Chemie Int Ed*, 2015).
2. Zheng *et al.* provided *in vivo* evidence showing that G4s form in response to DNA negative supercoiling. They observed that negative supercoiling efficiently triggers G4 formation and that the degree of G4 formation inversely correlates with the magnitude of supercoiling. These findings support our observation of a persistent high-FRET population under G4-destabilizing conditions, which we attribute to supercoiling-induced stabilization of G4 structures in our system (Zheng, K.-w. *et al.*, *Nucleic Acids Research*, 2020).

⇒ **Our observations and explanation:**

Under G4-destabilizing conditions, G4 structures in supercoiled DNA exhibit shorter lifetimes and tend to transition to DNA-only or R-loop states, as seen in the time traces presented in Supplementary Figure 11. This behavior contrasts with G4 formation in the presence of KCl (Figure 5b and 6a), where the G4 structures are long-lasting and stable. The high-FRET population observed in supercoiled DNA under G4-destabilizing conditions indicates that supercoiling provides additional stabilization to transiently formed G4 structures, even in the absence of monovalent cations.

⇒ This combined evidence from previous studies and our current results suggest that transcription in supercoiled DNA, even in G4-destabilizing conditions, can promote and stabilize G4 formation. The persistence of the high-FRET population under these conditions is therefore likely a consequence of supercoiling-induced stabilization of G4 structures.

Page 8, Line 257: "Based on the result, the stable high FRET state likely encompasses both R-loop and G4 (G4/R-loop)." This statement appears to contradict the earlier assignment in which the high FRET state was exclusively attributed to G4. The implications of the observed effects significantly change depending on which interpretation is assumed. Can the authors clarify?

⇒ We want to clarify that in FRET1 construct, both dyes are labeled on the non-template strand, which is only sensitive to the structure forming on the non-template strand (Fig. 4a and 5a). Hence, removing R-loop in template does not change the FRET value once the G4 forms stably (Fig. 5f). In our previous work, we had demonstrated that R-loop remains steadily alongside with G4 structure during transcription, which we defined G4/R-loop.

Pages 8-9: Most of the statements about Figure 5 are based on sample traces and have not been quantified in a way that justifies the conclusions. Such statements are often weak as they demonstrate significant variations among different traces. The authors should consider performing more quantitative analysis for this section.

⇒ We thank you for raising the importance of quantitative analysis. We would like to clarify that Figures 5f, 5h, 6e, and 5g indeed present quantified data, as detailed in the Methods section. Specifically, each FRET histogram in Figures 5f, 5h, and 6e reflects compiled FRET efficiency values from approximately 6,000 molecules across 15 to 20 movies, while Figure 5g shows the quantification of transcription events from over 230 molecules across 3 to 5 movies. The representative traces in Figures 5b and 5c are included to provide visual context; however, the conclusions are based on the quantified data shown in Figures 5f, 5h, 6e, and 5g.

⇒ To address potential concerns about variability among individual traces, we included Supplementary Figure 18, which displays additional traces and the intensity vs. FRET distribution for relaxed and supercoiled [FRET1] constructs. The 2D plot clearly reveals the mid-FRET population with enhanced intensity, corresponding to short-lived R-loops.

Supplementary Fig. 18: Representative single-molecule traces and intensity vs. FRET distribution for relaxed and supercoiled [FRET1] constructs.

(a and b) Representative single-molecule time traces of RNAP transcription for relaxed (a) and supercoiled (b) [FRET1] DNA constructs. Cy3 and Cy5 signals are shown in green and red, respectively. The dashed gray line indicates the time point when RNAP is introduced into the channel. In the relaxed [FRET1] construct (a), no FRET changes are observed, but short-lived

PIFE signals are present. In the supercoiled [FRET1] construct (b), FRET transitions are observed, accompanied by short-lived PIFE peaks. **(c and d)** Intensity vs. FRET distributions for accumulated traces of relaxed (c) and supercoiled (d) [FRET1] DNA constructs. The distributions are based on more than 50 molecules; exceeding this number results in dot density that obscures the clean distribution. In the relaxed construct (c), only populations with increased intensity and unchanged FRET values are observed. In contrast, the supercoiled construct (d) shows a high-FRET population (~0.9) with enhanced intensity and a mid-FRET population (~0.4–0.7) with increased intensity compared to the relaxed construct.

Page 10, Line 328: "In conclusion, our investigation highlights the critical role of G4 and R-loop structures in regulating transcription initiation dynamics through modulation of DNA superhelicity at the promoter region." As all constructs used in this study had PQS downstream of TSS, I am not sure whether their effects could still be considered as "transcription initiation".

⇒ We would like to clarify that the formation of G4 and R-loop structures downstream of TSS can modulate the transcription initiation by affecting local DNA superhelicity, which propagates to upstream promoter region. This is possible due to the transmission of torsional stress along the DNA molecule during transcription. Here, we applied two FRET constructs, FRET 1 and FRET 2, to monitor formation of the non-template structure and transcription initiation, respectively. These constructs had been verified in our previous work (Koh H. et al., 2018; Lee C.-Y et al., 2020).

Fig. 1b-c: How do the control measurement compare in linear and supercoiled configurations without normalization?

⇒ At the early stage, the supercoiled construct exhibits a higher transcription rate, as negative supercoiling enhances the RNAP loading rate. While the control construct does not contain PQS and thus lacks G4 formation, it still forms R-loops, as confirmed in Supplementary Figure 1 through S9.6 EMSA measurements, as well as in Figures 6d and 6g. However, the kinetics of R-loop formation in the control construct are slower than those in the NT-PQS (sp30) construct, eventually leading to a decrease in the transcription rate over time. Within the ~20-minute observation window for transcription measurement, the supercoiled DNA maintains a higher transcription rate compared to linear DNA when PQS is absent from the construct.

Fig.1. Dashed lines in this figure appear solid on my screen for this figure.

⇒ We apologize for any confusion caused by the incorrect annotation. We have updated the figure legend to reflect the correct information, ensuring that the data is now more clearly and accurately presented.

"line779: c) PQS orientation effect on RNA production. PQS locates 30 bp downstream of Transcription start site (TSS) either in non-template (NT-PQS) or in template (T-PQS). As a control, random sequences are inserted (Control). (b) Linearized plasmid with NT- PQS (left, blue line) shows an enhanced RNA production rate, while T-PQS (left, black line) shows a decreased RNA production rate compared to Control (left, gray line). The plasmid with NT-PQS (right, blue line) does not exhibit the enhanced RNA production compared to Control (right, gray line) and T-PQS (right, black line)."

Fig. 1e: Error bars are quite large. Are the differences significant? Have the authors performed any statistical test to establish the significance of the differences?

⇒ We appreciate the reviewer for raising this point regarding statistical significance. In our initial experiments with three repeats, the differences were not statistically significant, although consistent trends were observed across individual measurements. To address this concern, we performed additional experiments, increasing the sample size to five repeats. While the results showed slight variations compared to our initial observations, the overall trend remains consistent.

a

b

⇒ To ensure clarity, we have included (a) the statistical analysis results, including p-values, in the revised dataset and (b) the transcription rate data from the experiment with five individual repeats.

⇒ Due to space limitations in the main figure, statistical markers have not been added directly; however, the data, including p-values and raw data, are provided as a supplementary source file.

Figure 2c: The transcription rate for NT-sp30 in Fig. 1e is essentially same as the control (within the experimental error bars). However, substantial shifts are observed for this construct after transcription in Fig. 2c. This would suggest those shifts are not the cause of the transcription suppression. Given this, can the authors justify the following statement in Page 5, Line 161: "Based on the higher suppression seen in NT-sp10, the shifted bands likely correspond to DNA conformations that suppress transcription." Does the shift pattern in the control and that of sp30 look similar?

⇒ We appreciate this comment. First, in the supplementary figure 1, we have demonstrated that the shift pattern of control and sp30 are different. Second, we want to note that in the absence of superhelicity, PQS-NT shows higher transcription rate than the control (Fig 1b), but the enhancement is diminished in the presence of superhelicity (Fig 1e, sp30) and even worse in other positions. Therefore, the band shift is correlated with the transcription inhibition. In addition, the gel shift pattern cannot reflect the G4 formation and the stability of G4/R-loop, which are the concepts we expanded in the single molecule experiments.

Figure 2d: How did the authors quantify the shift to relaxed state?

⇒ We appreciate the opportunity to clarify the quantification method, as we missed including it in the manuscript. In response to your question, we have included this quantification method in the Methods section:

⇒ *"To quantify the shift from supercoiled to relaxed states as shown in Figure 2d and 2e, we obtained gel band intensities from the gel images using Fiji. After importing the gel images into Fiji, we measured the intensity of each band to determine the distribution of gel bands. For further analysis of topological shifts, we used OriginPro 2018 to fit a Gaussian function to the supercoiled state. The area under the Gaussian curve represented the intensity of the supercoiled bands. We then calculated the relaxed fraction as follows:*

$$Relaxed (\%) = \left(\frac{Total - supercoiled}{Total} \right) \times 100$$

where "Total" represents the total area in the gel band distribution, and "Supercoiled" refers to the area under the Gaussian curve fit for the supercoiled bands (Supplementary Figure 16)."

⇒ In addition, we added Supplementary Figure 16 to further clarify the quantification.

Figure 3a-b: The linear part of the curves (until about 20 min) appear to overlap, which would have suggested the same rate; however, the rates in (b) are not the same. Is there a criterion used to determine which part of the curve is treated as linear?

⇒ In this experiment, as the beacon continuously binds the transcript and is gradually depleted, we anticipated that the binding rate of the beacon would reduce over time. To address this, we set the linear region as the initial 20-minute interval, where linear fitting was feasible. We applied the same criteria used in our previous study (Lee C.-Y. et al., Nat Commun, 2020) to maintain consistency. Additionally, to illustrate the distinct transcription activity within the first 20 minutes, we have added a graph showing the initial 20-minute interval of Figure 3a as Supplementary Figure 20. This approach allowed us to effectively compare the transcription rates across different constructs during the early phase.

Figure 3b: Was the control construct supercoiled in Fig. 3b? If yes, shouldn't topoisomerase also resolve negative supercoiling in this construct and give rise to enhanced transcription rate?

⇒ Yes, the control construct in Figure 3b was supercoiled. Negative supercoiling impacts transcription in two ways: it increases the loading rate of RNAP by making the promoter region more accessible, and it also promotes G4/R-loop formation, which can influence transcription dynamics. We confirmed the presence of R-loops in the supercoiled control construct through

S9.6 EMSA experiments (Supplementary Figure 1) and single-molecule measurements (Figure 6). However, in the control construct, the R-loop effect is less significant because G4 formation, which would further stabilize the R-loop, is not supported. Therefore, the transcription rate with topoisomerase I reflects a combined effect of decreased RNAP loading due to reduced negative supercoiling and limited R-loop dynamics. This balance of effects results in the observed transcription output, where the difference in transcription rate for the control construct in Figure 3b is not as pronounced as in the PQS construct.

Figure 5f: How many molecules were included in the histogram in Fig. 5f? The differences between first and second panels in Fig. 5f appears negligible.

- ⇒ As described in the Methods section under 'Statistical Analysis for smFRET,' each FRET histogram in Figures 5f, 5h, and 6e was compiled from approximately 6,000 molecules, obtained from 15 to 20 separate movies.
- ⇒ Regarding the similarity between the first and second panels in Figure 5f, we note that the mid-FRET peak (~ 0.7), observed in individual time traces, has a very short lifetime before transitioning to the high-FRET G4 state. As described in the main text, *'The 0.7 FRET peak is not distinct because the R-looped state is highly transient in supercoiled DNA, as shown in Figure 5c-e'*, this brief duration means that the mid-FRET state does not accumulate prominently in the histogram, resulting in an apparent lack of difference between the two panels. This transient R-loop formation briefly creates a locally underwound state, lowering the energy barrier for PQS to fold into G4, as shown in Figure 5c and Supplementary Figures 7, and 18b and 18d.
- ⇒ However, under G4-destabilizing conditions in Fig. 5f, we observe a clear mid-FRET peak in the histogram (third panel), which disappears upon RNase H1 treatment (fourth panel), confirming that the mid-FRET state corresponds to the R-loop structure.

Figure 5 caption: The authors wrote: "The expected E of R-loop and G4 is ~ 0.7 and ~ 0.9 , respectively." How has this been established? Also, why is the entire range between 0.3-0.8 marked as R-loop if the expected range for it is around $E \sim 0.7$?

- ⇒ The FRET analysis for the R-loop and G4 was established in our previously (Lee et al., 2020). The FRET peak corresponding to R-loop is assigned from the disappearance after RNase H treatment, and the FRET peak for G4 is defined by pre-folding the structure during annealing process. The broad range of R-loop FRET value is due to the unistructural state of the non-template strand, where the G4 has not formed yet.
- ⇒ To clarify this, we've revised the manuscript as follows:
"line 253: Additionally, the R-loop exhibited a broad range of FRET values, reflecting the transient and fluctuating nature of R-loop formation."

Figure 5d caption: The authors wrote: "d) A closer examination of the light gray area of (c) reveals a distinct temporal shift between PIFE and FRET peaks." First, I do not see a light gray area in (c). Also, have the authors performed any analysis to justify this statement (other than observing it in a particular trace)? The time difference between the two events is very small. In fact, the onset of the events appears the same within time resolution of the experiment, but the donor

emission appears to reach its maximum value before the FRET change reaches its maximum.

⇒ We apologize for the labeling oversight. The figure legend should refer to the 'dashed black box' rather than a light gray area, marking the region where we observe the temporal shift between PIFE and FRET peaks. We have corrected this in the manuscript.

"line852: A closer examination of the dashed black box in (c) shows a distinct temporal shift, with the FRET peak emerging shortly after the PIFE peak."

⇒ Due to the short lifetime of this state, it is challenging to observe in the accumulated histogram, which represents the overall FRET behavior. To provide a clearer view, we have added 2D distribution plots for intensity and FRET values in Supplementary Figure 15c and d, illustrating both relaxed and supercoiled [FRET1] constructs. With these 2D plots, we can clearly distinguish the transient mid-FRET peak, which appears only in the supercoiled [FRET1] construct. This supports our conclusion that transiently formed R-loops facilitate stable G4/R-loop formation.

Figure 6 caption. I assume (c) is for -sp30 and (d) is for control. The authors should clarify.

⇒ We have revised the caption as follows:

"line870: (c-f) Single-molecule traces are synchronized either at the time of RNAP and NTP addition for NT-PQS (-sp30) (c) and control (d) or the time of forming long-lived G4 and R-loop structures for NT-PQS (-sp30) (e) and control (f)."

Figure 7a caption: The authors wrote: "...featuring Cy3 and Cy5 positioned 4 bp (-4) and 19 bp (+19) downstream from the TSS, respectively." Shouldn't "(-4)" be "(+4)"?

⇒ This is correct; the figure legend should indicate that

"line885: ..featuring Cy3 and Cy5 positioned 4 bp upstream (-4) and 19 bp downstream (+19) from the TSS, respectively"

We have revised the caption to accurately reflect this positioning.

Reviewer #2 (Remarks to the Author):

In this study, Jihee Hwang, et.al used single-molecule fluorescence and other techniques to study the interrelationships among RNA polymerase, promoter, G4, R-loop, and DNA supercoiling, and found that the formation of G4/R-loop changes the helical state of promoter DNA, thereby affecting the binding ability of RNA polymerase. This study reveals a possible way in which gene transcription is affected by G4/R-loop, and provides very meaningful information for our understanding of the function of PQS downstream of gene promoters. However, there is still room for improvement in some details, as described below.

1. In the transcription experiment of supercoiled plasmid, we did not see the authors describe how long the T7 RNAP transcription region is. If there is no terminator, T7 RNAP may transcribe the entire plasmid DNA, resulting in a global change in the helical state of the plasmid. Should the authors limit the length of T7 transcription to the region related to transcription initiation, so as to study the effect of R-loop formation in the promoter region on the change of the local helical state and the binding ability of T7 RNAP?

⇒ We included the *rrnB* T1 terminator in our construct, as it was shown to function effectively with T7 RNA polymerase in the study by Song et al. (Genes Cells, 2001). For the *sp30* construct, the distance from the T7 promoter to the terminator is 941 bp, while the total plasmid length is 3682 bp. This is essential information, and we have revised the Methods section to reflect it as follows:

"line466: To generate PQS-inserted plasmids with varying PQS positions relative to the transcription start site (TSS), we constructed a plasmid containing a GFP-expressing gene with PQS, the T7 promoter, and the rrnB T1 terminator (Supplementary Table 1). For the sp30 construct, the transcription region spans 941 bp from the T7 promoter to the rrnB T1 terminator within the total 3682 bp plasmid."

⇒ As shown in Supplementary Figure 12, we measured the full transcript length to be approximately 1200 bp, though this measurement may not be entirely precise due to the thickness of the band and the challenge of fully denaturing the long transcript on a UREA gel. However, this length provides a close estimate of the total transcript size. These results indicate that T7 RNA polymerase dissociates at the terminator site, preventing interference with subsequent rounds of transcription initiation. Therefore, the presence of the terminator ensures that local helical state changes, influenced by R-loop formation, are confined to the intended transcription region, allowing us to accurately study their effect on transcription initiation dynamics.

2. In line 111, the author claimed that "PQS promotes transcription in linear DNA but suppresses it in supercoiled DNA". however, the results in Figure 1b and 1c did not show PQS suppresses transcription in supercoiled plasmid.

⇒ While the results in Figures 1b and 1c may not show pronounced suppression within the initial time window used to measure the transcription rate, extending the observation period reveals a more significant difference.

- ⇒ Suppression occurs earlier in sp30 than in the control, likely due to superhelicity relaxation in the promoter region facilitated by G4-associated R-loops.
- ⇒ This observation is consistent with the gradual formation of G4 and/or R-loop structures over time, as well as transcriptional suppression observed in our single-molecule assays and simulations, as shown in Figures 7c and 7d.
- ⇒ The main comparison in our study is between **linear** vs. **supercoiled** DNA and our primary conclusion is that PQS in supercoiled DNA suppresses transcription compared to PQS in the linear DNA which enhances transcription.

3. There is no PQS sequence labeled in any of the result figures. What type of G-quadruplex can it form? Are there multiple conformations?

⇒ We thank you for bringing this to our attention. We used the same PQS sequence, c-MYC (5'-GGGTGGGTAGGG-3'), as in our previous study (Lee et al., *Nat Commun*, 2020). In our earlier studies (Kreig, A et al., *Nucleic Acids Res* (2015); Kim M et al., *Nucleic Acids Res* (2016)), we established that this sequence has a high GQ-forming propensity and predominantly adopts a **parallel** conformation in double-stranded DNA. Based on these findings, we expect the c-MYC in our current study to similarly form a stable parallel GQ conformation.

⇒ We realized that Supplementary Table 1, which contains construct information, was not attached. We have added it to the supplementary materials and updated the Methods section: *"line466: To generate PQS-inserted plasmids with varying PQS positions relative to the transcription start site (TSS), we constructed a plasmid containing a GFP-expressing gene with the PQS sequence, c-MYC (5'-GGGTGGGTAGGG-3'), the T7 promoter, and the rrnB T1 terminator (Supplementary Table 1)"*

4. PQS has a more obvious inhibition on transcription only when it is within 20 bases of the T7 promoter (Figure 1e). Is it possible that the formation of G4 and R-loop in the aborted transcription region causes the T7 promoter to partially unwind and thus cannot be bound by T7 RNAP?

⇒ Our previous work (Tang G.-Q. et al., *PNAS*, 2009) demonstrated that once the transcript length exceeds 12 nucleotides, the transcription process moves beyond the abortive phase, allowing T7 RNAP to form a stable elongation complex that efficiently translocates along DNA at approximately 100 nt/second. Since the PQS in sp10 is positioned within the first 12 nucleotides, it may have a more pronounced inhibitory effect by impacting the transcription initiation phase.

⇒ However, if this suppression were solely due to the proximity of PQS to the T7 promoter, we would not observe greater suppression in the supercoiled construct compared to the linear one. The stronger transcriptional suppression seen in supercoiled sp10, sp30, and sp60 constructs suggests that supercoiling amplifies the inhibitory effect of PQS, regardless of sequence positioning.

5. In Supplementary Figure 13, the amount and size of R-loops in NT-sp10 and NT-sp30 are similar, but in Figure 2b, the proportion of relaxed states in NT-sp10 is higher than that in NT-sp30. How can this be explained?

⇒ This is because the R-loops were measured after approximately 60 minutes, when the proportions of relaxed states in NT-sp10 and NT-sp30 become more similar as seen in Figure 2f.

6. Lines 321-323, Supplementary Figure 12 were conducted under G4 destabilizing condition. How do the authors conclude that "The level of negative supercoiling is likely reduced upon the formation of G4/R-loop structures to a level similar to that of a relaxed DNA".

⇒ We thank you for bringing this to our attention. In Supplementary Figure 12, the experiments conducted using the **relaxed DNA construct**, we observed approximately three transcription events before the negative supercoiling began to impede further transcription. Based on these observations, we concluded that the level of negative supercoiling is likely reduced upon the formation of G4/R-loop structures to a level similar to that of relaxed DNA.

⇒ To avoid confusion, we have revised the figure caption to accurately reflect the experimental conditions and findings.

"Supplementary Fig. 12: Transcription-generated negative supercoiling enhances the transcription rate by making the promoter region more accessible in the relaxed construct. The left panel quantifies the number of FRET peaks observed within 200 seconds after RNAP injection at time points 0 min, 10 min, and 30 min for NT-PQS and Control constructs (left y-axis, shown in blue and gray bars, respectively) in relaxed [FRET2]. Data are based on 3-4 replicates with over 200 molecules. The right schematic illustrates how multiple rounds of transcription increase RNAP loading rate by making the promoter more accessible. In the presence of G4 formation, NT-PQS may relieve negative supercoiling upon forming G4/R-loop structures."

7. Figure 5g shows that the transcription frequency of supercoiled plasmids is 10 times higher than that of linear and relaxed plasmids. However, in the Beacon Assays in Figures 1 and 3, the transcription efficiencies of different types of plasmids are not that different. It seems that the results of different experiments are inconsistent?

⇒ In Figure 5g, the transcription events captured reflect the **very initial phase** before the formation of a stable G4/R-loop structure. This early phase shows an enhanced transcription rate driven by pre-existing negative supercoiling, which stimulates G4/R-loop formation through both the increased transcription activity (Liu, J. et al., *Angewandte Chemie Int Ed*, 2015) and the underwound/loosened state of DNA base pairing (Zheng, K.-w. et al., *Nucleic Acids Research*, 2020; Stolz, R. et al. *Proc National Acad Sci*, 2019). Figure 7c represents

reduced transcription rate after the stable G4/R-loop structure forms ($E \sim 0.9$) and absorbs negative superhelicity,

⇒ This dual approach allows us to observe how transcription rate decreases as G4/R-loop formation increases, as seen in Figure 7c. Consequently, the transcription rates obtained from the beacon assay in Figures 1 and 3 should be considered alongside Figure 7 to provide a comprehensive view of transcriptional dynamics under varying supercoiling conditions.

8. The FRET1 model in Figure 5a may require further confirmation. As described in a previous study (Liu JQ, Xiao S, Hao YH, Tan Z. Strand-Biased Formation of G-Quadruplexes in DNA Duplexes Transcribed with T7 RNA Polymerase. *Angew Chem Int Ed Engl.* 2015 Jul 27;54(31):8992-6.), when PQS is located on the non-template DNA strand, G-quadruplex formation begins when the 3' end of RNA synthesis is approximately 7 nt away from PQS.

⇒ In our study, the presence of an R-loop means that the DNA strands are already separated, allowing G4 formation to occur in an open conformation.

⇒ This difference implies that, unlike in Liu et al., where transcription bubble proximity plays a significant role, our system allows G4 to form independently in the single-stranded region created by the R-loop. Therefore, we believe our FRET-based single molecule assay accurately captures the dynamics of R-loop-mediated G4 formation as depicted in Figure 5a.

9. As mentioned in question 3, PQS may form G-quadruplexes with different conformations. Is it possible that the phenomenon of mid FRET and high FRET is caused by the difference in G-quadruplex conformation?

⇒ In our previous study, we identified mid and high FRET as representing R-loop and G-quadruplex, respectively (Lee et al, *Nature comm*s, 2020). In addition, in our previous work (Kreig, A. et al., *Nucleic Acids Res*, 2015), we quantified the G4-forming propensity and conformation specificity of the c-MYC PQS in double-stranded DNA using single-molecule FRET. This study demonstrated that the c-MYC sequence predominantly forms a stable parallel G4 structure.

⇒ In light of your question, we have updated the main manuscript to emphasize that once the high-FRET state is reached, it remains stable, with no detectable conformational shifts within the G4.

"line249: The persistence of these high-FRET states indicates that no conformational change occurs within the G4."

10. In Figure 5, the description of no M⁺ is inaccurate. In fact, each NTP in the transcription reaction usually brings 2-3 monovalent cations, and the storage buffer of T7 RNA polymerase also brings a certain concentration of monovalent cations.

⇒ We thank you for your accurate observation regarding the presence of monovalent cations introduced through NTPs and the T7 RNA polymerase storage buffer. As noted in studies by Lane et al. (*Nucleic Acids Res*, 2008) and Mergny et al. (*Oligonucleotides*, 2003), G4 formation is particularly sensitive to potassium ions (KCl), which significantly enhance G4

stability. In our description, 'no M+' specifically indicates the omission of KCl from the reaction buffer, in contrast to conditions where KCl was included to support G4 stability.

- ⇒ In the manuscript, we updated this section to clarify the experimental setup as follows:
"line 246: Once it transitions to high-FRET, the state remains stable. The FRET histogram analysis revealed that the high FRET peak diminished under G4-destabilizing conditions by omitting KCl from the reaction buffer (Figure 5f, no M+), confirming the long-lived high-FRET as a stably folded G4 state (Figure 5f)."

Reviewer #3 (Remarks to the Author):

Manuscript by Hwang et al applied ensemble fluorescence, single molecule FRET-PIFE and gel-based assay, reports the cross-regulation effect of transcription, negative supercoiling and G4/R-loop formation. Detailed understanding of these events is crucial for the proper contextual placement of in vitro transcription experiments within the general framework of gene expression in cellular milieu.

Specific comments:

1). In the first part of the results, authors used "linear" DNA as control compared to "supercoiled" circular DNA to conclude the DNA topology affects transcription. Would the authors provide a set of measurement use "nicked" circular DNA to interpretate this effect? With the nicked site more downstream of the transcription which can release the supercoiling generated by transcription while maintain the circular form of the DNA.

⇒ We thank you for your thoughtful suggestion. We have tested nicked circular DNA, but it exhibited the same behavior as linear DNA, except in the case of sp10. Our linear DNAs were generated by cutting 10 base pairs upstream of the T7 promoter using EcoRI. This configuration likely prevents the accumulation of supercoiling changes.

2). Would author provide the detail of the "Beacon" specific binding place in the overall sequence, is it near the TSS or more downstream of the transcription sequence? Because DNA topology also influence the transcription processivity.

⇒ The beacon binding sequence is same as what we used in our previous work (Lee et al, 2020 and Lee et al, 2023). It is positioned 193 bp downstream from the TSS, chosen specifically to monitor transcription without directly affecting the initial steps at the promoter region. While DNA topology can indeed influence transcription processivity, placing the beacon further downstream minimizes any potential impact on transcription initiation, allowing us to focus on transcription elongation. We have updated the beacon sequence information, now specified to be in the middle of the GFP gene, in Supplementary Table 1.

3). In line 126 that "we obtained ~30%-40% enhanced mRNA production rate in PQS-NT compared to control and PQS-T, respectively", the authors would say "~30% and ~40%" or say "compared to both control and PQS-T".

⇒ We thank you for your question. To clarify, the enhancement in mRNA production rate for PQS-NT is approximately 30% compared to the control and approximately 40% compared to PQS-T. I will revise the text for clarity.

"line125: For linear DNA, consistent with our previous findings, we observed an enhanced mRNA production rate in PQS-NT, with an increase of approximately 30% compared to the control and about 40% compared to PQS-T, respectively (Figure 1b and 1c)."

4). Would the authors be more specific on how to calculate the "transcription rate (%)" in Figure 1"? By looking at the Figure 1, the increase in of fluorescence intensity at initial time and the final saturation plateau are both different, given that not only the transcription speed and the final mRNA amount are both different.

⇒ We thank you for your question. We observed consistent results with extended measurement times, where the transcription rate differences became more pronounced. The

extended observation period allowed us to capture a more significant transcriptional effect. We have attached the results from the extended time window for your review.

5). Line 141, I assumed the authors want to say "downstream of TSS in supercoiled plasmid" because the linear DNA showed no difference in transcription rate.

⇒ Sentence was revised.

6). Line 153, the authors choose NT-sp10 and NT-sp30 based on high and low level of suppression, would the author do one more test on mid-level of suppression such as NT-sp50 or NT-sp60? Because based on figure 1e, the suppression of the PQS to TSS position is not simple increase or decrease, but two phase. Do the authors have any hypothesis why this happen?

⇒ We have not measured the topological shift for NT-sp50 and NT-sp60. However, our measurements for NT-sp10 to NT-sp35 revealed the highest position dependency, including mid-level suppression observed at NT-sp20 and NT-sp25, as shown in Figure 2d.

⇒ The observed position-dependent transcriptional suppression is likely influenced by R-loop formation dynamics. Studies by Roy et al. (2009) and Belotserkovskii et al. (2022) suggest that G-clusters near PQS regions facilitate R-loop initiation, particularly when PQS is proximal to the transcription start site (TSS). Shorter RNA transcripts formed near the TSS make R-loop formation more favorable, leading to stronger transcriptional suppression. As RNA lengthens downstream, R-loop formation becomes less likely, reducing suppression. Interestingly, suppression increases again at distal PQS positions, suggesting a biphasic pattern linked to R-loop dynamics. Additionally, PQS in linear DNA was observed to increase transcription rates. This finding highlights the potential differences in how PQS impacts transcription depending on structural and positional contexts. These results underscore the complexity of transcriptional regulation mediated by PQS and R-loop dynamics, warranting further investigation to fully understand the mechanisms driving these effects.

⇒ We have incorporated a new discussion section to address this point:

"line430: Additionally, this study identifies a position-dependent effect on transcriptional suppression influenced by R-loop formation dynamics. Recent studies (Belotserkovskii et al., 2022; Roy et al.) suggest that G-rich clusters within PQS regions near the transcription start site (TSS) facilitate the initiation of transcription-induced R-loop formation³⁷, with R-loops forming more readily when PQS is positioned close to the TSS⁵³. When PQS is located

further downstream, the increased RNA transcript length decreases the likelihood of R-loop initiation and stabilization⁶⁵, which may explain why transcriptional suppression is strongest near the TSS and alleviates as PQS moves downstream (Figure 2). This biphasic pattern likely reflects the initial dynamics of R-loop formation, yet it warrants further investigation. Our results corroborate this, as we observed that PQS sequences close to the TSS (NT-sp10, 15, 20) exhibit the highest transcription suppression, with a gradual reduction as PQS is positioned further downstream (NT-sp25, 30) before the suppressive effect increases again at more distal positions (NT-sp35, 45, 60). This pattern underscores that the suppression effect is not linear but instead shows a biphasic dependence on PQS distance from the TSS, potentially linked to the dynamics of R-loop initiation.⁹). In supplementary Fig2b, how did the authors normalize the distribution?

7). Would the authors explain how they define/calculate the "shift to relax state" in figure 2?

Figure 2d: How did the authors quantify the shift to relaxed state?

⇒ We appreciate the opportunity to clarify the quantification method, as we missed including it in the manuscript. In response to your question, we have included this quantification method in the Methods section:

"line504: To quantify the shift from supercoiled to relaxed states as shown in Figure 2d and 2e, we obtained gel band intensities from the gel images using Fiji. After importing the gel images into Fiji, we measured the intensity of each band to determine the distribution of gel bands. For further analysis of topological shifts, we used OriginPro 2018 to fit a Gaussian function to the supercoiled state. The area under the Gaussian curve represented the intensity of the supercoiled bands. We then calculated the relaxed fraction as follows:

$$Relaxed (\%) = \left(\frac{Total - supercoiled}{Total} \right) \times 100$$

where "Total" represents the total area in the gel band distribution, and "Supercoiled" refers to the area under the Gaussian curve fit for the supercoiled bands (Supplementary Figure 16).

⇒ In addition, we added Supplementary Figure 16 to further clarify the quantification.

8). In line 175-176, the authors said "highest to lowest R-loops are detected for NT-sp10, -sp15, -sp20, -sp30, -sp35 (Figure 2d) which is inversely related to the transcription rate (Figure 1e)". It is correct except -sp30 and -sp35 since there is a decrease trend in Figure 1e.

⇒ We have incorporated a new discussion section to address this point:

"line430: Additionally, this study identifies a position-dependent effect on transcriptional suppression influenced by R-loop formation dynamics. Recent studies (Belotserkovskii et al., 2022; Roy et al.) suggest that G-rich clusters within PQS regions near the transcription start site (TSS) facilitate the initiation of transcription-induced R-loop formation³⁷, with R-loops forming more readily when PQS is positioned close to the TSS⁵³. When PQS is located further downstream, the increased RNA transcript length decreases the likelihood of R-loop initiation and stabilization⁶⁵, which may explain why transcriptional suppression is strongest near the TSS and alleviates as PQS moves downstream (Figure 2). This biphasic pattern

likely reflects the initial dynamics of R-loop formation, yet it warrants further investigation. Our results corroborate this, as we observed that PQS sequences close to the TSS (NT-sp10, 15, 20) exhibit the highest transcription suppression, with a gradual reduction as PQS is positioned further downstream (NT-sp25, 30) before the suppressive effect increases again at more distal positions (NT-sp35, 45, 60). This pattern underscores that the suppression effect is not linear but instead shows a biphasic dependence on PQS distance from the TSS, potentially linked to the dynamics of R-loop initiation.⁹) In supplementary Fig2b, how did the authors normalize the distribution?

9). In supplementary Fig2b, how did the authors normalize the distribution?

⇒ We apologize for the oversight. The data in supplementary Fig. 2b were not normalized. While analyzing the area of the shift, we normalized the band intensity by dividing it by the total area of the band intensity. However, in this figure, the y-axis should represent arbitrary units of gel band intensity. We have corrected the labeling accordingly and sincerely appreciate you bringing this to our attention.

10). Line 198, authors said that "pre-treating DNA with Topo 1 showed a dose-dependent increase in transcription in the NT-sp10 and NT-sp30". It shouldn't be described as dose dependent since it's a factor of efficiency, if treat sufficient time all supercoiling will be released. Also, the authors should be how they quench the Topo I reaction.

⇒ As detailed in the Methods section, we controlled the degree of DNA relaxation by incubating plasmids with varying units of Topo I (10, 5, 3.3, and 2.5 units) for 30 minutes at 37 °C, followed by purification using the QIAquick PCR Purification Kit (QIAGEN). This approach allowed us to create a gradient of superhelicity levels, rather than relying solely on enzyme concentration for relaxation.

⇒ Additionally, we confirmed the gradual relaxation achieved in these constructs through gel analysis, as shown in Figure 3a and Supplementary Figure 4, which provide a visual representation of the different superhelicity levels. Thank you for the opportunity to clarify these points.

⇒ We revised the line 198 to the follows:

"line196: We investigated the relaxation effect by treating the DNA with topoisomerase I (Topo I) prior to transcription (Figure 3a and Supplementary Figure 4) and by adding Topo I during transcription. For pre-treatment, plasmids were incubated with varying units of Topo I (10, 5, 3.3, and 2.5 units) for 30 minutes at 37 °C, followed by purification to control the degree of relaxation. Gel analysis confirmed the resulting gradient of superhelicity levels, as shown in Supplementary Figure 4."

11). Would author specify the Topo I concentration in certain treat time in the figure 3 instead of highest-high-mid-low-no?

⇒ We've revised the legend for Figure 3 to improve clarity:

"line809: Plasmid-NT-sp10, NT-sp30, and control samples were pre-treated with topoisomerase I (Topo I) at different concentrations to create varying degrees of

superhelicity. The gel image on the left shows NT-sp10 treated with 10 units (highest), 5 units (high), 3.3 units (mid), and 2.5 units (low) of Topo I, representing a gradient of supercoiling levels. For further details, see Supplementary Figure 4 and the Methods section."

12). Line 216-217, would the authors mark the lane with number in figure 4b left, otherwise the lane 1 in the figure is a ladder.

⇒ We thank you for your suggestion. We have added lane numbers above each gel lane in Figure 4b to clearly differentiate the samples and avoid confusion with the ladder.

13). Fig 4b, the lane of relaxed DNA seems still supercoiled, but with less superhelicity.

⇒ To achieve a completely relaxed state, we would need to treat the DNA with Topoisomerase I after the ligation step. However, taking this step would significantly reduce the yield of labeled DNA, particularly as the final yield of the [FRET2] construct is already around 30%. Further treatment with Topoisomerase I would lower this yield to less than 10%, complicating data collection.

⇒ Despite this, our current relaxed DNA construct serves well for our experiments, as it sufficiently represents the behavior of relaxed DNA. This is supported by the transcription events observed in Figure 5g, where the transcription rates for linear and relaxed DNA appear similar, indicating that our construct adequately models the relaxed state for this study.

14). In the [FRET1] strategy, the author indicate that the mid-FRET state is $E \sim 0.7$, in figure 5f and e, the R-loop state is (blue box) is span from $E \sim 0.4$ to $E \sim 0.8$ stage made it confusing. By looking at the traces in supplementary fig7, it is easy to understand the transition is a procedure and it progress fast and made the FRET efficiency looks spanning a wide range. I suggest the author to better clarify in the text.

⇒ We thank you for your insightful feedback. As you noted, the broad range of R-loop FRET values arises from the transient and fluctuating nature of the R-loop formation process,

particularly in supercoiled DNA. In response to your suggestion, we have revised the manuscript accordingly to clarify this point.

"line 253: Additionally, the R-loop exhibited a broad range of FRET values, reflecting the transient and fluctuating nature of R-loop formation."

15). Line 289, will it be the Figure 6c, d, bottom rather than the top?

⇒ To enhance clarity, we have separated the labels as c-f rather than using 'top' and 'bottom' for panels c and d.

16). Line 317, "ie., 200 sec, 10 and 30 minutes" however the figure 7c marked as "0, 10 and 30 mins".

⇒ Thank you for noting the labeling in Figure 7c. We measured transcription events over a 200-second interval starting from the time points marked as 0, 10, and 30 minutes. To clarify this process, we will revise the text to indicate that events were recorded within a 200-second window at each of these intervals.

⇒ To clarify this process, we have revised the manuscript as follows:

"line 321: To test this expected sequence of events, we quantified the FRET peaks at various stages of transcription over a 200-second interval at each time point—specifically at 0, 10, and 30 minutes after the addition of RNAP and NTP (Figure 7c)."

17). Would the author explain better how to get the supercoiling density in figure 7d, rather than just cite the paper?

⇒ Thank you for requesting clarification on how the supercoiling density in Figure 7d was obtained. We have revised as follows:

"line329: In addition, we employed an event-driven stochastic simulation to model supercoiling density changes in our system.⁵⁸ This simulation, as detailed in Tripathi et al. (2021), tracks the effects of co-transcribing RNA polymerases on DNA topology, specifically how transcription-induced supercoiling accumulates and influences the promoter region over time.⁵⁸ Through this model, we observed that as G4/R-loops (or R-loops in the control) accumulate, the level of promoter supercoiling progressively increases. Notably, the NT-PQS construct shows a more rapid increase in supercoiling at the promoter compared to the control, as depicted in Figure 7d and Supplementary Figures 14 and 15. This model allows us to directly link the transcription-driven supercoiling dynamics with the regulatory effects of G4/R-loop structures, providing insight into the interplay between supercoiling and transcription regulation."

18). The supplementary information missing supplementary table 1.

⇒ We apologize for the oversight and will add it to the supplementary materials to ensure all referenced information is available.

19). Line 363, "linearized PQS showed only a long-lived R-loop and subsequent G4 formation, which infrequently occurred (3/138) with very slow kinetics." Here the authors want to describe the "linearized PQS only has the scenario of long-lived R-loop" or "the linearized PQS also has short-lived R-loop, but only the long-lived R-loop form G4"?

⇒ For linear DNA, we observe only long-lived R-loops that subsequently lead to G4 formation, albeit infrequently and with slower kinetics (as shown in Supplementary Figure 8, with only 3 occurrences out of 138). This indicates that, in the linear construct, short-lived R-loops do not appear, and only the long-lived R-loops contribute to stable G4 formation. Thus, in our linear DNA observations, we are describing a scenario where only long-lived R-loops drive G4 formation.

In contrast, supercoiled DNA shows a markedly different behavior. Here, multiple rounds of short-lived R-loops are observed, driven by transcription-generated negative supercoiling and the underwound state of DNA base pairing. These transiently formed R-loops provide strand separation, which is sufficient to facilitate G4 formation without the requirement for long-lived R-loop stabilization, as seen in the linear construct

⇒ To clarify the meaning based on your feedback, we have revised the manuscript as follows:

"line374: Contrary to the supercoiled DNA, linearized PQS only forms a long-lived R-loop and subsequent G4 formation, which infrequently occurred (3/138) with very slow kinetics (Supplementary Figure 8)."

20). Line 374, the authors said supercoiled PQS construct form 100% G4/R-loop structure, but

according to fig 6e, only 60% formed G4/R-loop. Would the authors clarify this?

⇒ We thank you for pointing this out. In the supercoiled PQS construct, we observed only short-lived R-loops, which quickly transitioned to stable G4/R-loop structures. This means that we did not detect any long-lived R-loops in the supercoiled state without accompanying G4 stabilization. Therefore, our statement of '100% G4/R-loop formation' in the supercoiled construct reflects the fact that all observed R-loops in this context rapidly stabilized as G4 structures.

⇒ To clarify the meaning based on your feedback, we have revised the manuscript as follows:
"line383: In addition, the supercoiled PQS construct exhibited only short-lived R-loops that quickly transitioned to G4-stabilized R-loops, resulting in 100% of detected R-loops being associated with stable G4 formation. In contrast, the R-loops formed in the non-PQS construct were significantly less stable (Figure 6c)."

21). Line 496, did the authors want to say "EtBr" rather than EDTA?

⇒ We thank you for pointing this out. It should indeed be 'EtBr' (ethidium bromide) instead of 'EDTA.' I have updated the text accordingly.

22). Line 509, the buffer recipe in the bracket looks like "imaging buffer", the authors might mark it after the "transcription buffer" by mistake.

⇒ We thank you for your correction. We have revised as follows:
"line544: The imaging buffer was prepared freshly by mixing transcription buffer with an oxygen scavenging system (1mg/ml glucose oxidase, 0.8 % v/v glucose, ~2mM Trolox, and 0.03 mg/ml catalase)."

23). Would the authors plot the Fig 2f and Fig 3e in a quantitative way?

⇒ While it would be ideal to present these figures quantitatively, the values we measured are relative and serve to illustrate trends rather than precise quantities. Therefore, it is challenging to display them in a fully quantitative manner. To avoid potential confusion and maintain clarity, we have decided to remove these schematic figures from the final version. We appreciate your feedback in helping us streamline the presentation.

24). In the legend of figure 5(d) state "the light grey area of (c)", but there is no grey area in the (c). And the color box "blue box" and "red box" "grey box" in the figure is not easy distinct, would the authors edit it better?

⇒ We apologize for the labeling oversight. The caption should refer to the 'dashed black box' rather than a light gray area, marking the region where we observe the temporal shift between PIFE and FRET peaks. We have corrected this in the manuscript.

"line854: (d) A closer examination of the dashed black box on the left side of (c) reveals a distinct temporal shift between PIFE and FRET peaks, with a FRET peak emerging shortly after a PIFE peak. (e) The dashed black box on the right side of (c) shows that after several transitions, an additional FRET change from ~0.6 to ~0.9 is observed."

⇒ Following your suggestion, we have also updated the colors of the boxes, changing blue to orange for the R-loop and red to blue for the stable G4/R-loop to align with the color scheme

used in our previous study (Lee C.-Y. et al., *Nat Commun*, 2020).